# The Intriguing Effects of Substituents in the *N*-Phenethyl Moiety of Norhydromorphone: A Bifunctional Opioid from a Set of “Tail Wags Dog” Experiments

**DOI:** 10.3390/molecules25112640

**Published:** 2020-06-06

**Authors:** Meining Wang, Thomas C. Irvin, Christine A. Herdman, Ramsey D. Hanna, Sergio A. Hassan, Yong-Sok Lee, Sophia Kaska, Rachel Saylor Crowley, Thomas E. Prisinzano, Sarah L. Withey, Carol A. Paronis, Jack Bergman, Saadet Inan, Ellen B. Geller, Martin W. Adler, Theresa A. Kopajtic, Jonathan L. Katz, Aaron M. Chadderdon, John R. Traynor, Arthur E. Jacobson, Kenner C. Rice

**Affiliations:** 1Department of Health and Human Services, Drug Design and Synthesis Section, Molecular Targets and Medications Discovery Branch, Intramural Research Program, National Institute on Drug Abuse and the National Institute on Alcohol Abuse and Alcoholism, National Institutes of Health, 9800 Medical Center Drive, Bethesda, MD 20892-3373, USA; wangmeining8911@163.com (M.W.); Tom.irvin8@gmail.com (T.C.I.); caherdman@gmail.com (C.A.H.); Ramsey.hanna1@gmail.com (R.D.H.); 2Department of Health and Human Services, Center for Molecular Modeling, Office of Intramural Research, Center for Information Technology, National Institutes of Health, Bethesda, MD 20892, USA; hassan@mail.nih.gov (S.A.H.); leeys@mail.nih.gov (Y.-S.L.); 3Department of Medicinal Chemistry, School of Pharmacy, University of Kansas, Lawrence, KS 66045-7582, USA; sophia.kaska@uky.edu (S.K.); crowleyrachels@gmail.com (R.S.C.); prisinzano@uky.edu (T.E.P.); 4Behavioral Biology Program, McLean Hospital/Harvard Medical School, 115 Mill Street, Belmont, MA 02478, USA; swithey@mclean.harvard.edu (S.L.W.); cparonis@mclean.harvard.edu (C.A.P.); jbergman@hms.harvard.edu (J.B.); 5Center for Substance Abuse Research, Lewis Katz School of Medicine of Temple University, 3500 N. Broad St., Philadelphia, PA 19140, USA; sinan@temple.edu (S.I.); gellerellen@gmail.com (E.B.G.); baldeagl@temple.edu (M.W.A.); 6Department of Health and Human Services, Psychobiology Section, Molecular Neuropsychiatry Research Branch, Intramural Research Program, National Institute on Drug Abuse, National Institutes of Health, Baltimore, MD 21224, USA; theresa.kopajtic@gmail.com (T.A.K.); jkatzzz@gmail.com (J.L.K.); 7Department of Pharmacology and Edward F Domino Research Center, University of Michigan Medical School, Ann Arbor, MI 48109, USA; Aaron.Chadderdon@gmail.com (A.M.C.); jtraynor@umich.edu (J.R.T.)

**Keywords:** opioid, bifunctional ligands, (−)-*N*-phenethylnorhydromorphone analogs, [^35^S]GTPgammaS assay, forskolin-induced cAMP accumulation assays, β-arrestin recruitment assays, MOR and DOR agonists, respiratory depression, bias factor, molecular modeling & simulation

## Abstract

(−)-*N*-Phenethyl analogs of optically pure *N*-norhydromorphone were synthesized and pharmacologically evaluated in several in vitro assays (opioid receptor binding, stimulation of [^35^S]GTPγS binding, forskolin-induced cAMP accumulation assay, and MOR-mediated β-arrestin recruitment assays). “Body” and “tail” interactions with opioid receptors (a subset of Portoghese’s message-address theory) were used for molecular modeling and simulations, where the “address” can be considered the “body” of the hydromorphone molecule and the “message” delivered by the substituent (tail) on the aromatic ring of the *N*-phenethyl moiety. One compound, *N*-*p*-chloro-phenethynorhydromorphone ((7a*R*,12b*S*)-3-(4-chlorophenethyl)-9-hydroxy-2,3,4,4a,5,6-hexahydro-1*H*-4,12-methanobenzofuro[3,2-e]isoquinolin-7(7a*H*)-one, **2i**), was found to have nanomolar binding affinity at MOR and DOR. It was a potent partial agonist at MOR and a full potent agonist at DOR with a δ/μ potency ratio of 1.2 in the ([^35^S]GTPγS) assay. Bifunctional opioids that interact with MOR and DOR, the latter as agonists or antagonists, have been reported to have fewer side-effects than MOR agonists. The *p*-chlorophenethyl compound **2i** was evaluated for its effect on respiration in both mice and squirrel monkeys. Compound **2i** did not depress respiration (using normal air) in mice or squirrel monkeys. However, under conditions of hypercapnia (using air mixed with 5% CO_2_), respiration was depressed in squirrel monkeys.

## 1. Introduction

It is well-known that various *N*-substituents in the classical opioid-type of structure (e.g., 4,5-epoxymorphinans, morphinans, 6,7-benzomorphans, 5-phenylmorphans) can change prototypical *N*-methyl substituted agonist opioids to opioid antagonists. Most familiar, replacement of the *N*-methyl with an *N*-allyl moiety in morphine and oxymorphone converted them to the antagonists nalorphine and naloxone (Scheme 1).

Exactly how the *N*-substituent interaction with amino acid residues in the receptor induces that change remains uncertain. The *N*-methyl substituent in morphine and oxymorphone either permits or does not prevent the ligand from receptor interactions that result in analgesia and the well-known panoply of opioid side-effects. Concurrent with their interaction with G-proteins, potent clinically utilized opioids also recruit β-arrestin2. This recruitment has been hypothesized to underly the undesirable effects of opioids, including respiratory depression, inhibited gastrointestinal transport, and tolerance [1,2,3,4], although recent data using β-arrestin2 knockout mice cast doubt on that hypothesis [4].

Replacement of an *N*-methyl group with a different substituent, the *N*-phenethyl moiety, has also been shown to change an opioid agonist to an antagonist in at least one of the types of classical opioids; however, that outcome is not consistent for all opioid structures. Most *N*-phenethyl-substituted opioids, such as *N*-phenethylnormorphine [5], *N*-phenethylnoroxymorphone [6], and 2′-hydroxy-5,9-dimethyl-*N*-phenethylbenzomorphan [6] have been found to be potent μ-opioid receptor (MOR) agonists, but *N*-phenethyl-5-phenylmorphan acts as a MOR antagonist [7]. In that 5-phenylmorphan series, we found that the agonist vs antagonist activity of the compound was dependent on chirality. The 1*R*,5*R*,9*S*-enantiomer of the *N*-*p*-nitrophenethyl-5-phenylmorphan was found to be a potent MOR agonist, and the 1*S*,5*S*,9*R*-enantiomer acted as a MOR antagonist [8].

*N*-Phenethylnorhydromorphone (**S11**, Appendix A) was first synthesized in 2001 by Lo and McErlane in 1.3% yield; it was not further examined at that time because their synthesis did not provide the quantity of **S11** necessary for in vitro or in vivo assays [9]. It could be anticipated that it, like *N*-phenethylnormorphine and all other classical opioids, would display the side-effects displayed by morphine. We obtained **S11** through an improved synthesis (see Appendix A), and we found that **S11** acts as a bifunctional ligand, with a DOR-MOR δ/μ ratio = 39 in a functional assay (Table 2). It displayed extremely high ΜOR potency (EC_50_ = 0.04 nM) compared to morphine (EC_50_ = 4.7 nM), DOR potency (EC_50_ = 1.54 nM), and also κ-opioid receptor (KOR) activity (EC_50_ = 23 nM). This interesting pharmacology prompted our use of **S11** as a template for modifying its activity with the addition of different substituents on the nitrogen atom. We hypothesized that relatively small changes in, for example, the *N*-phenethyl group, hereafter referred to as the “tail” of the molecule, e.g., by adding substituents in various positions on the aromatic ring (Table 1), might modify the ligand’s activity and, perhaps, enhance its therapeutic potential. This “tail wags dog” experiment can be considered a subset of the message-address concept [10] as applied to opioids, where the “address” can be considered the bulk of the hydromorphone molecule and the “message” delivered by the substituent on the aromatic ring of the *N*-phenethyl moiety. We needed to modulate (decrease) the agonist potency of the **S11** compound at MOR and KOR without disturbing its DOR activity. Since we were unaware of a theoretical model that could predict which substituent would be able to tweak the molecule in that way, we decided to begin our exploration with the design and synthesis and evaluation of compounds with varied substituents on the nitrogen atom and on the aromatic ring of the phenethyl moiety, followed by rationalization of their activity using molecular modeling and dynamics simulations.

A bifunctional compound that acted as a MOR agonist and DOR antagonist would be of the greatest interest, since that combination of receptor interactions is believed to produce antinociceptive activity with fewer of the unfortunate side-effects of opioids [11,12,13]. The receptor interaction induced by ligands with MOR agonist and DOR agonist activity would also be of interest because that type of combination of MOR and DOR occupancy has also been noted to produce fewer of the undesirable side-effects caused by MOR agonist opioids [14,15,16]. For example, Su et al., have found that one of the DOR functions was to modulate or counteract the respiratory depression caused by the MOR and, most importantly, further noted that both DOR agonists and DOR antagonists acted similarly in counteracting respiratory depression [17]. It has been found that deaths due to opioid overdose are directly related to an opioid’s effect on respiratory depression [18]. It is theoretically possible that a potent agonist with the “correct” δ/μ potency ratio would have diminished effects on respiration or not depress respiration at all. The question of what that “correct” δ/μ ratio might be has not as yet been answered. Our previous attempts to determine this in the 5-phenylmorphan series failed due to their lack of potency [19,20]. There have been many reports of lessened side-effects induced by bifunctional ligands, mostly reduced gastrointestinal effects, a few equivocal results on reduced tolerance and dependence, and other reports have indicated reduced respiratory depression [21,22,23,24,25,26,27].

## 2. Results and Discussion

### 2.1. Chemistry

To prepare the necessary quantity of *N*-norhydromophone base (**S10** in Appendix A) we used a previously reported efficient process (hydromorphone synthesis, Appendix A) for the conversion of (−)-tetrahydrothebaine (**S1**) to dihydromorphine (**S3**) [28]. We found that using the HCl salt of compound **S3** as the reactant instead of the hydrate (product of the basic hydrolysis of **S2**) could dramatically increase the yield of the oxidation [29], achieving hydromorphone hydrochloride (**S4**) in 90% yield. Transformation of hydromorphone **S4** into one of our key intermediates, **S7**, was performed in four steps [30]. Optically pure (−)-*N*-substituted hydromorphone analogs with different substituents on the *N*-phenethyl moiety were obtained by *N*-alkylation of the secondary amine **S10** (Scheme 2).

### 2.2. In Vitro Studies

We prepared four groups of compounds (Scheme 2); two cyano analogs (**1a**, **1b**) and two aromatic ring compounds without substituents (**1c**, **1d**) in group 1, two *N*-nitrophenethyl compounds (**2a**, **2b**) in group 2, *N*-*p*-methylphenethyl and *p*-methoxyphenethyl compounds (**2c**, **2d**) in group 3, and three fluorophenethyl-substituted compounds (**2e**, **2f**, **2g**), a *p*-trifluoromethylphenethyl (**2h**), and a *p*-chlorophenethyl (**2i**) compound in group 4. Opioid receptor binding data (*K*i, nM) and stimulation of ([^35^S]GTPγS) binding were obtained for all of these compounds (Table 1). Two additional compounds were added to group 4 in Table 2 (forskolin-induced cAMP accumulation assay), a 4-bromo (**2j**) and a 2,6-dichlorophenethyl compound (**2k**), and two additional compounds were added to group 1, hydromorphone (**S5**) and *N*-phenethylnorhydromorphone (**S11**). These four compounds were only examined using the forskolin-induced cAMP accumulation assay. The ability of all of the compounds to recruit β-arrestin was also determined (Table 2).

Opioid receptor binding data (*K*i, nM) and stimulation of ([^35^S]GTPγS) binding were obtained for all of these compounds (Table 1). Two additional compounds were added to group 4 in Table 2 (forskolin-induced cAMP accumulation assay), a 4-bromo (**2j**) and a 2,6-dichlorophenethyl compound (**2k**), and two additional compounds were added to group 1, hydromorphone (**S5**) and *N*-phenethylnorhydromorphone (**S11**). These four compounds were only examined using the forskolin-induced cAMP accumulation assay. The ability of all of the compounds to recruit β-arrestin was also determined (Table 2).

#### 2.2.1. Opioid Receptor Binding, Ligand Efficacy and Potency ([^35^S]GTPγS Binding Assays)

The MOR binding affinity, in the receptor binding assay, of *N*-cyanomethyl (**1a**) and *N*-cyanopropyl compounds (**1b**) in group 1 (Table 1), both without an aromatic ring on the nitrogen atom, were comparable to the affinity of morphine (*Ki* = 4.51 and 4.25 vs 3.26 for morphine). However, **1a** had only partial MOR agonist activity (47% stimulation) and very low potency (EC_50_ = 425 nM) in the [^35^S]GTPγS assay (Table 1), and **1b**, in contrast, was a MOR antagonist in that assay (antagonist activity was assumed given the high MOR binding affinity, and lack of [^35^S]GTPγS stimulation at MOR). All of the compounds in group 1 (Table 1) that were assessed at DOR had relatively low receptor binding affinity (*Ki* > 70 nM). At KOR, **1**a had a *Ki* of about 90 nM, whereas **1b** had a higher binding affinity at KOR than MOR, a > 30-fold increase in KOR agonist affinity due to the extension of the carbon chain from *N*-cyanomethyl to *N*-cyanopropyl. Compounds in group 1 with an extended and rotationally restricted *N*-substituent (compound **1c**) or a bulky N-substituent (compound **1d**) showed little binding affinity or potency at any opioid receptor. Compounds were considered inactive and were not tested (NT) in functional assays when they were found to displace <50% of the radioligand at a 100 nM concentration in an exploratory binding assay (not described in the Material and Methods Section).

The *m*-(**2b**) and *p*-*N*-nitrophenethylnorhydromorphone (**2a**) in group 2 in Table 1 showed a >10-fold difference in MOR binding affinity in the receptor binding assay and a remarkable change from potent MOR agonist **2a** to MOR antagonist **2b** (Table 1) was observed in the [^35^S]GTPγS assay, apparently induced by the change in the position of the aromatic ring’s substituent. This bolstered our hypothesis that we could influence and considerably alter activity by substitution on the phenyl ring.

The group 3 alkyl and alkoxy compounds had very high MOR affinity (K*i* < 1 nM) and high DOR affinity (K*i* = 5–6 nM) in the receptor binding assay, and while **2c** was a potent partial MOR agonist in the [^35^S]GTPγS assay, the methoxy compound **2d** appeared to be a MOR antagonist in that assay. Both of these compounds had EC_50_ < 35 nM at DOR with **2c** acting as a full agonist (95% stimulation) and **2d** a partial agonist (49% stimulation). Compound **2d** was also a potent KOR agonist (EC50 = 5.9 nM), although it was not efficacious at KOR (21.8% stimulation).

The halides in group 4 (Table 1) harbored the most interesting compound **2i**, from the perspective of having a desirable δ/μ potency ratio. All of the halides had high affinity at MOR and DOR (K*i* ranged from 0.3 to 2.7 nM at MOR and 4 to 16 nM at DOR), and less affinity at KOR (K*i* > 20 nM), in the receptor binding assays. Additionally, all group 4 compounds had nanomolar MOR potency in the [^35^S]GTPγS assay (EC50 = 2.0–3.4 nM) and all except **2i** and **2h** had lower DOR agonist potency (EC50 > 50 nM). The trifluoromethyl compound **2h** had moderate DOR potency (EC_50_ = 36 nM), whereas **2i** had nanomolar potency at DOR (EC50 = 2.4 nM), with a δ/μ potency ratio of 1.2.

#### 2.2.2. Ligand Potency and Efficacy Using the Forskolin-induced cAMP Accumulation Assay

As seen in the forskolin-induced cAMP accumulation assay (group 1, Table 2), **1a** had morphine-like potency, as it did in the [^35^S]GTPγS assay. In contrast, compounds **1a**, **1c** and **1d** had relatively low potency for DOR or KOR cAMP stimulation.

Again, as in the [^35^S]GTPγS assay, **1c** with restricted rotation and **1d** with a bulky side-chain were less potent than the cyanomethyl compound **1a**. The standard compounds for comparison purposes, **S5** and **S11**, hydromophone and *N*-phenethylnorhydromorphone in group 1, were relatively potent at MOR (EC_50_ = 1.67 and 0.04 nM, respectively) and **S11** was potent at DOR (EC_50_ = 1.54 nM) and somewhat less potent at KOR (EC_50_ = 22.7 nM). The parent compound hydromorphone (**S5**) was essentially inactive at DOR and KOR (EC_50_ > 130 nM). The *N*-phenethylnorhydromorphone (**S11**), had an EC_50_ δ/μ ratio = 38.5, and that was possibly too high for the mitigation of side-effects that might be provided by interaction with DOR.

The bias factor of all of the tested compounds ranged between ca. 0.4 and 3.5 (Table 2). Examination of the MOR β-arrestin recruitment for **S11** and **2i** indicated that they both had a lower bias factor than morphine (indicating their greater ability to recruit β-arrestin). All of these G-protein biased ligands recruit β-arrestin almost as well, or better than morphine. If recruitment of β-arrestin correlated with the side-effects of these compounds, they should all cause, for example, respiratory depression, as well as other side-effects caused by interaction with MOR. As noted previously, however, the ability of MOR ligands to recruit β-arrestin may not have any bearing on whether they will or will not display opioid-like side-effects [4].

The MOR antagonist profile of **2a** in group 2 (Table 1), the *p*-nitro compound, that was seen in the [^35^S]GTPγS assay was not observed in the forskolin-induced cAMP accumulation assay (Table 2). In the cAMP assay, both **2a** and **2b** had MOR agonist activity (EC_50_ = 0.05 and 5.2 nM, respectively); a >100-fold potency change due to a positional shift of an aromatic substituent.

In group 3 in Table 2, the alkyl and alkoxy compounds, **2c** and **2d**, were found to have relatively high potency (EC_50_ = 0.08 and 0.13 nM, respectively) at MOR and at DOR (EC_50_ = 1.0 and 2.7 nM, respectively. They were also KOR agonists (EC_50_ = 8.7 and 7.4 nM, respectively). In the cAMP assay (Table 2), **2d** was not found to have MOR antagonist activity.

In group 4 (Table 2), the most interesting compound was again found to be **2i.** It had extremely high agonist potency at MOR and DOR (EC_50_ = 0.05 and 0.53 nM, respectively), and was efficacious at both receptors. It was much less potent at KOR (EC_50_ = 55 nM). The *o*-fluoro compound **2g** was notable for its relatively high MOR agonist potency (0.01 nM), with full (101%) efficacy. The *p*-bromo compound **2j** also appeared to be of interest in that its EC_50_ δ/μ ratio = 6.5 was less than the ratio found for **2i** in the cAMP assay (δ/μ ratio = 10 for **2i**). Compound **2j** was also potent at both MOR and DOR in the cAMP assay (EC_50_ < 1 nM), but it had moderate agonist potency at KOR (20 nM). Although KOR agonists have therapeutic potential, they also have undesirable CNS-mediated side-effects (e.g., dysphoria, hallucinations) [32].

We hypothesized that a compound with a δ/μ potency ratio of less than 7 in the [^35^S]GTPγS assay would be desirable if the compound were to display other than the full array of undesirable effects that clinically used analgesics manifest. Recent work on bifunctional compounds explored compounds with δ/μ potency ratios of 5 to 7 in [^35^S]GTPγS assay assays, and found that those compounds had a less disruptive effect on locomotor activity than morphine or oxymorphone [33].

#### 2.2.3. Molecular Modeling and Simulations.

Specifically placed moieties at the tail end of the *N*-phenethyl substituent changed the compound from a MOR partial agonist in the [^35^S]GTPγS assay (e.g., *N*-*p*-nitrophenethyl-norhydromorphone, **2a**) to a MOR antagonist (e.g., *N*-*m*-nitrophenethylnorhydromorphone, **2b**). In the forskolin-induced cAMP accumulation assay, that same positional shift of the substituent in the aromatic ring in **2a** to **2b** caused a >100 fold change in potency. We used quantum chemical calculations and molecular dynamics simulations to determine if a moiety in precise positions on the aromatic ring in the *N*-phenethyl moiety of norhydromorphone would display sufficient differences in their interaction with MOR to induce the change in activity and/or potency. More generally, our simulations allowed us to identify the critical residues interacting with the body (Figure 1) and the tail (Figure 2) of the ligands that are responsible for the differences in receptor properties. Together, the experimental and simulation data led us to propose a set of general rules for the *N*-phenethyl-substitutions to impart specific behaviors, such as partial or full agonist or possible antagonist activities, which may help design compound with novel properties (see details in the Appendix A).

##### Body-Opioid Receptor (OR) interactions

All the compounds considered in our simulations have similar body-OR interaction patterns regardless of the substituents on the N-phenethyl ring; the interactions are consistent with those reported recently for a series of phenethyl oxymorphone compound bound to the active MOR [33]. These are shown in Figure 1 for both the MOR and the DOR, and involve polar/charged residues in transmembrane helix (TMH) 3, 5 and 6, and hydrophobic residues in TMH 5, 6, and 7. A contact is deemed significant if it persists for at least 50% of the time (see example in Appendix A), although not necessarily all the interactions are seen at a given time. Different substituents, however, lead to different statistics of the individual interactions due to modest repositioning of the ligands resulting from different tail-OR interactions. All the residues in direct contact with the body are conserved in both receptors. Three additional, non-conserved residues, each belonging to TMH 5, 6, and 7 interact indirectly with -O- and -OH groups of the body via short water chains. Thus, suitable substitutions that engage these residues more directly (e.g., polar or H-bond interactions) may help modulate the MOR and DOR activity, potency and affinities independently.

##### Tail-OR interactions

Because the tail is located deep inside the pocket, relatively small changes in the *N*-phenylethyl ring via substitutions perturb residues located in different regions of the ORs and engage different TMHs (Appendix A). The constrained environment of the tail suggests that these interactions can induce major changes in the OR structure and/or dynamics, as confirmed by preliminary data from principal component analysis (to be published). Unlike the relatively rigid ligand body, the tail has several energetically similar conformers (Appendix A). Some of these conformers can be ruled out based on unfavorable steric interactions (cf. Computational Section 5). However, the simulations show that all of the conformers selected by the pocket can be stabilized because each substituent can always find favorable interactions through polar/nonpolar or H-bond interactions. It is noted that none of these conformers lose the critical polar/H-bond interaction between the protonated nitrogen of the ligands and the carboxylate of the anchoring aspartic acid D147 (MOR) or D128 (DOR) (cf. Appendix A). The multiplicity of binding modes may explain, in part, the tendency of most ligands in the Table 1 series to display partial agonist activity, i.e., with some modes leading to agonist and others to antagonist activity. This scenario may coexist with the more traditional view of partial activity as the result of multiple substates of the receptor stabilized by a single conformer (not observed in the simulations). We carried out a comparative analysis of the experimental observations summarized in Appendix A. Here we focused on the results pertaining to the effects of F and Cl substitution at the *p-*position of the *N*-phenethyl moiety on both receptors (Figure 2A, additional details in SI). The *p*-F (**2e**) was found to be a potent partial MOR agonist and weak partial DOR agonist, whereas *p*-Cl (**2i**), although still a potent partial MOR agonist, becomes a potent full DOR agonist, an unexpected result with therapeutic potential.

One of the conformers of **2e** and of **2i** showed the same pattern of interactions with the DOR, engaging both TMH 6 (W274) and 7 (N310, S311, N314); this conformer (not shown) was expected to elicit the same dynamic behavior of the receptor and was unlikely to be responsible for the observed partial vs. full agonist activities. The second conformer did show qualitative differences (Appendix A); when compared to **2e**, **2i** partially disengaged TMH 7 and engaged TMH 3. On the one hand, Cl is larger than F, resulting in higher polarizability together with a longer C-Cl bond distance, and is less electronegative, resulting in a less negative partial charge and weaker electric field at the atom surface. These differences appeared to be enough for **2i** to interact more favorably with both S135 and S311, which were in opposite sides across the pocket; **2e** instead interacted more closely with the polar groups of adjacent N310 and N314 on the same TMH 7. In both cases, the ligands interacted with TMH 6, showing persistent interactions with the -NH group of W274 and with the H atoms on the F270 ring via weak polar interactions. Although halogen-bonding interactions were not included in the forcefield [34,35], it could be predicted that the incipient σ-hole and equatorial negative charge of the Cl atom (Appendix A) would further stabilize this pattern of interactions. There may be other qualitative differences between **2e** and **2i** if the latter interacts with the aromatic ring via C-Cl/π interactions [36]; in this case, the side-chain may adopt a different conformation and affect the dynamics of TMH 6. Despite the differences in the pattern of interactions, the type and frequency of polar and non-polar contacts were fundamentally the same in both ligands, which may explain their similar *Ki* values. When all the interactions were considered, only two residues of DOR show unique interactions with these ligands: N314 (only with **2e**) and S135 (only with **2i**). Therefore, substituents that interacted with S135 (or engaged TMH 3 near this residue) and interacted less strongly with N314 (or disengaged TMH 7) may confer potent full DOR agonism. The difference in atomic size, polarizability, and electronegativity, as well as the putative C-Cl/π interactions, appear to play a role in the difference between **2e** and **2i**. Accordingly, it would be of interest to see the effects of *p*-Br and *p*-I on DOR behavior. In MOR, the patterns of interactions of **2i** were similar to those in DOR (Figure 2B): one conformer interacted with TMH 6 and 7 and the other with TMH 3 and 7. This may be enough to impart potent agonism, as in the DOR, but only partial agonist activity in MOR. When all the interactions were considered, F289 is the only residue that was unique in the interaction patterns of **2i** with the receptors. The present investigation indicated that changing the balance of interactions of the phenethyl ring with S154, N332, and F289 by a suitable substituent may help design potent, full DOR agonists.

The constrained environment of the tail substituent on the aromatic ring of the *N*-phenethyl moiety located deep inside the receptor pocket suggested that these interactions can induce major changes in the OR structure or dynamics. The consequences of such effects were already observed in a previous study where a *p*-NO_2_ substitution in the ring elicited significant changes in OR activity and efficacy [8]; computational studies of other GPCRs have also shown the importance of the lower strata of the binding pocket to affect function [37]. Thus, moieties in specific positions on the phenyl ring in *N*-phenethylnorhydromorphone might convert a potent MOR agonist to MOR antagonist or significantly change its potency. This was shown using in vitro assays with the nitro substituent on the phenyl ring (in **2a** and **2b**, Table 1). Depending on the position of the nitro substituent in the phenyl ring, one of the compounds (with a *p*-nitro substituent, **2a**) was a potent low efficacy MOR agonist with subnanomolar affinity and the other (with a *m*-nitro substituent, **2b**) was a high-affinity MOR antagonist in the [^35^S]GTPγS assay (Table 1), a minor positional change inducing a significant change in activity. In the forskolin-induced cAMP accumulation assay, a major difference in potency was observed with these compounds.

### 2.3. In Vivo Data

#### 2.3.1. Respiratory Depression Assays in Mice

The *p*-chlorophenyl compound **2i** was among the most interesting of the compounds in that it exhibited high MOR and DOR affinity and potency and was a potent efficacious DOR agonist and a partial MOR agonist in the [^35^S]GTPγS assay (Table 1). It was an exceptionally potent MOR agonist in the forskolin-induced cAMP accumulation assay (Table 2). Few compounds have been noted in the literature that combine potent MOR partial agonist and potent DOR full agonist activity in a δ/μ ratio of about 7 in opioid receptor binding studies and in a δ/μ ratio of about 1 from potency studies in the [^35^S]GTPγS assay (Table 1). We thought that it would be of interest to further examine that compound in vivo.

In mice, **2i** did not depress respiration rate in the presence of normal air. Figure 3A shows time courses of saline, morphine (10 mg/kg), and different doses of **2i** on respiration rate. Figure 3B shows the calculated area under the curve (AUCs) from 6 min to 45 min post injection. As seen in Figure 3B, 10 mg/kg morphine significantly reduced (*p* < 0.0001) respiration rate compared to saline (One-way ANOVA revealed a significant effect for treatment F(5,38) = 18.34, *p* < 0.0001).

The doses chosen were based on the squirrel monkey tail withdrawal latency assay and the highest dose (0.1 mg/kg) was about 5 or 6 times higher than the ED_50_ values at 50 and 52 °C from the tail withdrawal latency assay (the usual dose studied to observe side-effects is about 4× the ED50). Compound **2i** (0.01–0.1 mg/kg) had no effect on respiration rate in this assay in mice although morphine, as expected, significantly decreased respiratory rate. Results for oxygen saturation (SpO_2_) indicated that neither morphine nor **2i** had any effect on SpO_2._ from 6 min to 45 min post-injection (data not shown).

#### 2.3.2. Antinociceptive Studies and Respiratory Depression Studies in Squirrel Monkeys

Further studies evaluated the effects of **2**i and, for comparison, morphine, in assays of antinociception and respiratory depression in nonhuman primates. In a squirrel monkey tail withdrawal latency assay, the *p*-chlorophenethyl compound **2i** exhibited the observed partial MOR agonist in vitro characteristics (in the [^35^S]GTPγS assay) by having a full effect at moderate (50 °C) and hot (52 °C) but not at very hot (55 °C) water temperatures. In comparison, morphine elicited full antinociceptive effects at all three water temperatures. The compound **2i**, like morphine, also produced dose-related behavioral impairment, evident as decreases in operant performance and, consequently, the number of reinforcers obtained during an operant task interspersed between tail-withdrawal tests. The doses of **2i** that produced behavioral impairment were similar to those that produce antinociception in 50 °C and 52 °C water (Figure 4).

The ED_50_ doses for producing antinociception at 50 °C or 52 °C were, respectively, 0.010 and 0.015 mg/kg, and the ED50 dose for reducing the number of reinforcers earned (behavioral impairment), was 0.022 mg/kg, resulting in ED_50_ ratios (behavioral impairment/antinociception) that were about 2 or less. Ratio values greater than 1 imply some behavioral selectivity in observed effects although, as can be seen in Table 3, morphine had an even greater behavioral impairment/antinociception potency ratio than did **2i**. Further studies evaluated whether **2i** differed from morphine in its capacity to produce respiratory depression in squirrel monkeys.

The results from a respiratory depression assay in squirrel monkeys correlated somewhat with data obtained using mice. As shown in Figure 5 during exposure to air alone, neither **2i** (0.003–0.1 mg/kg) nor morphine (0.03–3.0 mg/kg) had effects on respiratory rate or overall ventilation (minute volume) in squirrel monkeys, whereas in mice morphine (10 mg/kg), but not **2i** (0.01–0.1 mg/kg), depressed ventilation. In contrast, during exposure to 5% CO_2_ mixed in air (hypercapnia), both **2i** and morphine significantly decreased ventilation in squirrel monkeys, resulting from respiratory depressant effects (Figure 5).

The ED_50_ values for producing antinociception in 52 °C water, decreasing the number of reinforcers earned in an operant behavioral task, and decreasing ventilation in 5% CO_2_, as well as calculated potency ratios across the procedures, are summarized in Table 3. Here, a complicated picture emerges in which morphine has a larger, and hence more favorable, potency ratio for antinociceptive and behaviorally disruptive effect than does **2i**, whereas **2i** has a higher ratio for behaviorally disruptive and respiratory depressant effects. Indeed, the potency ratio for morphine for behavioral disruption and respiratory depression was ≤ 1, indicating that breathing was decreased at similar doses to those that decreased behavior, whereas 2-fold higher doses of **2i** were needed to decrease CO_2_-stimulated breathing (Figure 5).

The ED_50_ values for producing antinociception in 52 °C water, decreasing the number of reinforcers earned in an operant behavioral task, and decreasing ventilation in 5% CO_2_, as well as calculated potency ratios across the procedures, are summarized in Table 3. Here, a complicated picture emerges in which morphine has a larger, and hence more favorable, potency ratio for antinociceptive and behaviorally disruptive effect than does **2i**, whereas **2i** has a higher ratio for behaviorally disruptive and respiratory depressant effects. Indeed, the potency ratio for morphine for behavioral disruption and respiratory depression was ≤1, indicating that breathing was decreased at similar doses to those that decreased behavior, whereas 2-fold higher doses of **2i** were needed to decrease CO_2_-stimulated breathing.

## 3. Material and Methods

### 3.1. General Information

Melting points were determined on a B-545 instrument (Büchi, Labortechnik AG, Flawii, Switzerland) and are uncorrected. Proton and carbon nuclear magnetic resonance (^1^H and ^13^C-NMR) spectra were recorded on a Gemini-400 spectrometer (Varian, Palo Alto, CA, USA) with the values given in ppm (TMS as internal standard) and *J* (Hz) assignments of ^1^H resonance coupling. Mass spectra (HRMS) were recorded on a VG 7070E spectrometer (VG Analytical Ltd., Altrincham, Cheshire, England, UK) or a SX102a mass spectrometer (JEOL, Tokyo, Japan) 41 polarimeter (PerkinElmer, Shelton, CT, USA) at room temperature. Gas chromatography (GC) was performed on a 6850 GC system (Agilent Technologies, Santa Clara, CA, USA) equipped with a VL MSD detector. Thin layer chromatography (TLC) analyses were carried out on prepackaged plates using various gradients of CHCl_3_/MeOH containing 1% of 28% NH_4_OH (CMA) or gradients of EtOAc:*n*-hexane. Visualization was accomplished under UV light or by staining in an iodine chamber. Flash column chromatography was performed using RediSep Rf normal phase silica gel cartridges. Atlantic Microlabs, Inc. (Norcross, GA, USA) or Micro-Analysis, Inc. (Wilmington, DE, USA) performed elemental analyses, and the results were within ± 0.4% of the theoretical values. All spectra were obtained on the free base.

### 3.2. Synthesis. General Procedure for Formation of Tertiary Amines.

*(7aR,12bS)-3-(4-Chlorophenethyl)-9-hydroxy-2,3,4,4a,5,6-hexahydro-1H-4,12-methanobenzofuro[3,2-e]-isoquinolin-7(7aH)-one* (**2i**): To a stirred solution of *N*-norhydromorphone (**S10** in the Appendix A, 0.406 g, 1.5 mmol) in DMF (10 mL) were added NaHCO_3_ (0.504 g, 6 mmol) and 1-(2-bromoethyl)-4-chlorobenzene (0.480 mL, 3.3 mmol). The mixture was heated at 90 °C overnight, cooled to room temperature, filtered through Celite, and concentrated in vacuo. Water (10 mL) was added and the mixture extracted with CHCl_3_ (3 × 20 mL). The combined organics were washed with H_2_O (5 × 20 mL), dried over Na_2_SO_4_, filtered through Celite, and concentrated in vacuo to afford **2i** free base as a crude colorless oil. Purification of this oil by SiO_2_ column chromatography with 10% NH_4_OH in MeOH/CHCl_3_ (gradient, 0 → 4% of 10% NH_4_OH) afforded **2i** (0.420 g, 70%) as a colorless oil. ^1^H-NMR (DMSO-d_6_): δ 9.09 (s, 1H), 7.30 (d, *J* = 8.4 Hz, 2H), 7.26 (d, *J* = 8.4 Hz, 2H), 6.51 (d, *J* = 8.0 Hz, 1H), 6.46 (d, *J* = 7.6 Hz, 1H), 4.78 (s, 1H), 3.17 (s, 1H), 2.79 (d, *J* = 18.4 Hz, 1H), 2.71–2.56 (m, 5H), 2.52 (dd, *J* = 14.4 Hz, 4.8 Hz, 1H), 2.48–2.40 (m, 1H), 2.23 (dd, *J* = 18.2 Hz, 5.4 Hz, 1H), 2.11 (d, *J* = 13.6 Hz, 1H), 2.04–1.93 (m, 2H), 1.76–1.71 (m, 1H), 1.47 (d, *J* = 10.0 Hz, 1H), 1.01–0.91 (m, 1H); ^13^C-NMR (DMSO-d_6_): δ 209.2, 144.3, 140.1, 139.6, 131.0, 130.8, 128.5, 127.9, 124.9, 119.6, 117.3, 90.7, 57.3, 56.4, 47.1, 44.8, 41.8, 40.1, 35.3, 33.4, 25.5, 21.2; HRMS (TOF MS ES^+^) calcd for C_24_H_25_ClNO_3_ (M + H^+^) 410.1523, found 410.1527. **2i HCl**: An HCl salt was prepared by dissolving **2i** free base in hot *i*-PrOH (5.0 mL) followed by the addition of 37% HCl (0.10 mL, 3 equiv). It was concentrated in vacuo to afford the salt. Anal. Calcd for C_24_H_24_ClNO_3._0.5H_2_O·0.5C_3_H_8_O (**2i HCl**·0.5H_2_O·0.5C_3_H_8_O): C, 63.29; H, 6.11; N, 2.77; found: C, 63.09; H, 6.23; N, 2.89%. ${\left[\alpha \right]}_{D}^{20}$ − 138.5 (*c* 0.65, MeOH, HCl·0.5H_2_O·0.5*i*-PrOH).

*2-((7aR,12bS)-9-Hydroxy-7-oxo-1,2,4,4a,5,6,7,7a-octahydro-3H-4,12-methanobenzofuro[3,2-e]isoquinolin-3-yl)acetonitrile* (**1a**): The general procedure with **S10** (0.406 g, 1.5 mmol), 2-bromoacetonitrile (0.230 mL, 3.3 mmol), NaHCO_3_ (0.504 g, 6 mmol), and DMF (10 mL) at room temperature. Purification by column chromatography afforded **1a** free base (0.250 g, 50%) as a white solid. ^1^H-NMR (CD_3_OD): *δ* 6.61 (d, *J* = 8.0 Hz, 1H), 6.56 (d, *J* = 8.4 Hz, 1H), 4.76 (s, 1H), 3.74 (d, *J* = 17.2 Hz, 1H), 3.63 (d, *J* = 17.2 Hz, 1H), 3.36 (dd, *J* = 5.2 Hz, 2.8 Hz, 1H), 2.99 (d, *J* = 18.4 Hz, 1H), 2.69 (dd, *J* = 11.6 Hz, 3.6 Hz, 1H), 2.62 (dt, *J* = 12.6 Hz, 3.4 Hz, 1H), 2.53 (td, *J* = 14.0 Hz, 4.4 Hz, 1H), 2.46–2.35 (m, 2H), 2.28 (dt, *J* = 14.0 Hz, 3.0 Hz, 1H), 2.13 (td, *J* = 12.4 Hz, 4.8 Hz, 1H), 1.89–1.83 (m, 1H), 1.72 (ddd, *J* = 12.4 Hz, 3.2 Hz, 1.6 Hz, 1H), 1.14 (ddd, *J* = 27.2 Hz, 13.2 Hz, 2.8 Hz, 1H); ^13^C-NMR (CD_3_OD): δ 210.0, 144.1, 139.4, 126.7, 124.4, 119.7, 117.4, 117.0, 90.9, 58.3, 46.7, 44.4, 42.2, 41.8, 39.2, 34.5, 25.1, 21.1; HRMS (TOF MS ES^+^) calcd for C_18_H_19_N_2_O_3_ [M + H]^+^ 311.1396, found 311.1397. Anal. Calcd for C_18_H_18_N_2_O_3_·0.1CHCl_3_·0.25H_2_O (**1a**·0.1CHCl_3_·0.25H_2_O): C, 66.48; H, 5.42; N, 8.46; found: C, 66.52; H, 5.74; N, 8.57%. ${\left[\alpha \right]}_{D}^{20}$ −190.5 (*c* 0.43, CHCl_3_/MeOH (20/1)).

*2-((7aR,12bS)-9-Hydroxy-7-oxo-1,2,4,4a,5,6,7,7a-octahydro-3H-4,12-methanobenzofuro[3,2-e]isoquinolin-3-yl)butanenitrile* (**1b**): The general procedure with **S10** (0.406 g, 1.5 mmol), 4-bromobutanenitrile (0.344 mL, 3.3 mmol), NaHCO_3_ (0.504 g, 6 mmol), and DMF (10 mL). Purification by column chromatography afforded **1b** free base (0.350 g, 70%) as a colorless oil. ^1^H-NMR (CD_3_OD): *δ* 6.59 (d, *J* = 8.0 Hz, 1H), 6.54 (d, *J* = 8.4 Hz, 1H), 4.74 (s, 1H), 3.22 (dd, *J* = 5.6 Hz, 2.8 Hz, 1H), 2.91 (d, *J* = 18.4 Hz, 1H), 2.72–2.48 (m, 7H), 2.37 (dd, *J* = 18.4 Hz, 5.6 Hz, 1H), 2.28 (dt, *J* = 13.6 Hz, 3.2 Hz, 1H), 2.19 (td, *J* = 12.0 Hz, 3.2 Hz, 1H), 2.09 (td, *J* = 12.0 Hz, 4.4 Hz, 1H), 1.88–1.77 (m, 3H), 1.66 (ddd, *J* = 12.4 Hz, 3.0 Hz, 1.8 Hz, 1H), 1.14 (ddd, *J* = 27.4 Hz, 13.0 Hz, 2.8 Hz, 1H); ^13^C-NMR (CD_3_OD) *δ* 210.4, 144.1, 139.1, 127.1, 125.0, 119.9, 119.6, 117.3, 91.0, 57.7, 52.8, 47.3, 44.5, 41.7, 39.4, 34.8, 25.3, 22.9, 20.8, 13.8; HRMS (TOF MS ES^+^) calcd for C_20_H_23_N_2_O_3_ [M + H]^+^ 339.1709, found 339.1711. An HCl salt was prepared by dissolving **1b** free base in hot *i*-PrOH (5.0 mL) followed by the addition of concentrated aq HCl (0.10 mL, 3 equiv) and cooling to 5 °C. The crystals were filtered and air-dried to give **1b** as its HCl salt. Anal. Calcd for C_20_H_22_N_2_O_3_·HCl·1.5H_2_O·0.25C_3_H_8_O (**1b**·HCl·1.5H_2_O·0.25C_3_H_8_O): C, 59.65; H, 6.74; N, 6.49; found: C, 59.78; H, 6.77; N, 6.72%. ${\left[\alpha \right]}_{D}^{20}$ − 188.3 (*c* 0.6, CHCl_3_/MeOH (1/10), HCl·1.5H_2_O·0.25C_3_H_8_O salt).

*(7aR,12bS)-3-Cinnamyl-9-hydroxy-2,3,4,4a,5,6-hexahydro-1H-4,12-methanobenzofuro[3,2-e]isoquinolin-7(7aH)-one (1c)* Thkye general procedure with **S10** (0.406 g, 1.5 mmol), (*E*)-(3-bromoprop-1-en-1-yl)benzene (0.650 mg, 3.3 mmol), NaHCO_3_ (0.504 g, 6 mmol), and DMF (10 mL). Purification by column chromatography and crystallization from MeOH afforded **1c** free base (0.180 g, 31%) as a white solid. ^1^H-NMR (DMSO-d_6_) *δ*. 9.10 (s, 1H), 7.41 (d, *J* = 7.6 Hz, 2H), 7.29 (t, *J* = 7.4 Hz, 2H), 7.20 (t, *J* = 7.4 Hz, 1H), 6.57–6.49 (m, 3H), 6.26 (dt, *J* = 15.6 Hz, 6.4 Hz, 1H), 4.80 (s, 1H), 3.32–3.28 (m, 1H), 3.21–3.14 (m, 2H), 2.86 (d, *J* = 18.4 Hz, 1H), 2.57–2.48 (m, 3H), 2.23 (dd, *J* = 18.2 Hz, 5.4 Hz, 1H), 2.12 (dt, *J* = 14.0 Hz, 2.8 Hz, 1H), 2.01 (d, *J* = 7.2 Hz, 2H), 1.77–1.72 (m, 1H), 1.48 (d, *J* = 9.2 Hz, 1H), 0.97 (ddd, *J* = 27.2 Hz, 12.8 Hz, 2.2 Hz, 1H); ^13^C-NMR (DMSO-d_6_): δ 209.2, 144.4, 139.7, 137.2, 131.8, 129.0, 128.5, 127.9, 127.8, 126.6, 124.9, 119.6, 117.3, 90.8, 57.2, 56.9, 47.1, 45.0, 41.9, 40.1, 35.3, 25.5, 20.9; HRMS (TOF MS ES^+^) calcd for C_25_H_26_NO_3_ [M + H]^+^ 388.1913, found 388.1906. Anal. Calcd for C_25_H_25_NO_3_·0.25H_2_O (**1c**·0.25H_2_O): C, 76.60; H, 6.56; N, 3.57; found: C, 76.28; H, 6.42; N, 3.62%. ${\left[\alpha \right]}_{D}^{20}$ − 214.0 (*c* 0.4, CHCl_3_/MeOH (1/10)).

*(7aR,12bS)-3-((2,3-Dihydro-1H-inden-2-yl)methyl)-9-hydroxy-2,3,4,4a,5,6-hexahydro-1H-4,12-methano-benzofuro[3,2-e]isoquinolin-7(7aH)-one* (**1d**): The general procedure with **S10** (0.406 g, 1.5 mmol), 2-(bromomethyl)-2,3-dihydro-1H-indene (0.470 mL, 3.3 mmol), NaHCO_3_ (0.504 g, 6 mmol), and DMF (10 mL). Purification column chromatography and crystallization from chloroform afforded **1d** (0.190 g, 32%) as white solid. ^1^H-NMR (CD_3_OD): δ 7.16–7.14 (m, 2H), 7.08–7.05 (m, 2H), 6.60 (d, *J* = 8.0 Hz, 1H), 6.55 (d, *J* = 8.0 Hz, 1H), 4.75 (s, 1H), 3.24 (dd, *J* = 5.0 Hz, 2.6 Hz, 1H), 3.07–2.99 (m, 2H), 2.93 (d, *J* = 18.4 Hz, 1H), 2.76–2.59 (m, 5H), 2.56–2.49 (m, 3H), 2.35–2.26 (m, 2H), 2.20 (td, *J* = 12.0 Hz, 3.0 Hz, 1H), 2.11 (td, *J* = 12.0 Hz, 4.2 Hz, 1H), 1.86–1.80 (m, 1H), 1.65 (d, *J* = 12.0 Hz, 1H), 1.13 (ddd, *J* = 27.2 Hz, 13.2 Hz, 2.2 Hz, 1H); ^13^C-NMR (CD_3_OD): δ 210.6, 144.1, 142.7, 142.6, 139.1, 127.2, 125.7, 125.2, 124.1, 119.6, 117.2, 91.1, 59.6, 57.5, 45.1, 41.8, 39.5, 37.2, 37.1, 37.0, 34.9, 25.4, 20.7; HRMS (TOF MS ES^+^) calcd for C_26_H_28_NO_3_ [M + H]^+^ 402.2069, found 402.2068. Anal. Calcd for C_26_H_27_NO_3_·CHCl_3_ (**1d**·CHCl_3_): C, 62.30; H, 5.12; N, 2.68; found: C, 62.26; H, 5.42; N, 2.69%. ${\left[\alpha \right]}_{D}^{20}$ −137.0 (c 0.77, CHCl_3_/MeOH (10/1)).

*(7aR,12bS)-9-Hydroxy-3-(4-nitrophenethyl)-2,3,4,4a,5,6-hexahydro-1H-4,12-methanobenzofuro[3,2-e]-isoquinolin-7(7aH)-one* (**2a**): The general procedure with **S10** (0.406 g, 1.5 mmol), 1-(2-bromoethyl)-4-nitrobenzene (759 mg, 3.3 mmol), NaHCO_3_ (0.504 g, 6 mmol), and DMF (10 mL). Purification by column chromatography afforded **2a** (0.070 g, 10%) as a colorless oil. ^1^H-NMR (CD_3_OD): δ 8.16 (d, *J* = 8.4 Hz, 2H), 7.51 (d, *J* = 8.4 Hz, 2H), 6.61 (d, *J* = 8.4 Hz, 1H), 6.56 (d, *J* = 8.0 Hz, 1H), 4.77 (s, 1H), 3.32–3.30 (m, 1H), 2.99–2.72 (m, 6H), 2.61–2.49 (m, 2H), 2.38 (dd, *J* = 18.6 Hz, 5.8 Hz, 1H), 2.29 (dt, *J* = 14.0 Hz, 3.0 Hz, 1H), 2.24 (td, *J* = 12.2 Hz, 3.4 Hz, 1H), 2.11 (td, *J* = 12.4 Hz, 4.8 Hz, 1H), 1.88–1.81 (m, 1H), 1.69 (d, *J* = 12.0 Hz, 1H), 1.15 (ddd, *J* = 27.2 Hz, 13.2 Hz, 2.0 Hz, 1H); ^13^C-NMR (100 MHz, CD_3_OD): δ 210.4, 148.6, 146.5, 144.1, 139.2, 129.5, 127.0, 124.9, 123.0, 119.6, 117.3, 91.0, 57.5, 55.6, 47.2, 44.7, 41.6, 39.4, 34.7, 33.4, 25.3, 20.6; HRMS (TOF MS ES^+^) calcd for C_24_H_25_N_2_O_5_ [M + H]^+^ 421.1763, found 421.1762. An HCl salt was prepared by dissolving **2a** free base in hot *i*-PrOH (5.0 mL) followed by the addition of concentrated aqueous HCl (0.10 mL, 3 equiv) and cooling to 5 °C. The crystals were filtered and air-dried to give **2a** as its HCl salt. Anal. Calcd for C_24_H_24_N_2_O_5_·HCl·1.75H_2_O (**2a**·HCl·1.75H_2_O): C, 58.99; H, 5.69; N, 5.56; found: C, 59.02; H, 5.88; N, 5.74%. ${\left[\alpha \right]}_{D}^{20}$ −107 (*c* 0.4, MeOH, HCl·1.75H_2_O salt).

*(7aR,12bS)-9-Hydroxy-3-(3-nitrophenethyl)-2,3,4,4a,5,6-hexahydro-1H-4,12-methanobenzofuro[3,2-e]-0isoquinolin-7(7aH)-one* (**2b**): The general procedure with **S10** (0.406 g, 1.5 mmol), 1-(2-bromoethyl)-3-nitrobenzene (759 mg, 3.3 mmol), NaHCO_3_ (0.504 g, 6 mmol), and DMF (10 mL). Purification by column chromatography afforded **2b** free base (0.035 g, 5%) as a colorless oil. ^1^H-NMR (CD_3_OD): *δ* 8.16 (t, *J* = 3.0 Hz, 1H), 8.05 (ddd, *J* = 8.0 Hz, 2.4 Hz, 0.8 Hz, 1H), 7.66 (d, *J* = 7.6 Hz, 1H), 7.51 (t, *J* = 7.8 Hz, 1H), 6.59 (d, *J* = 8.0 Hz, 1H), 6.55 (d, *J* = 8.0 Hz, 1H), 4.75 (s, 1H), 3.28–3.27 (m, 1H), 2.97–2.84 (m, 4H), 2.82–2.71 (m, 2H), 2.60–2.47 (m, 2H), 2.37 (dd, *J* = 18.6 Hz, 5.8 Hz, 1H), 2.30–2.21(m, 2H), 2.09 (td, *J* = 12.2 Hz, 4.6 Hz, 1H), 1.86–1.79 (m, 1H), 1.68 (ddd, *J* = 12.4 Hz, 3.2 Hz, 1.6 Hz, 1H), 1.13 (ddd, *J* = 27.4 Hz, 13.0 Hz, 2.8 Hz, 1H); ^13^C-NMR (CD_3_OD): *δ* 210.3, 148.2, 144.1, 142.8, 139.2, 134.9, 129.0, 127.0, 125.0, 123.2, 120.6, 119.6, 117.3, 91.0, 57.5, 55.7, 47.3, 44.7, 41.7, 39.4, 34.8, 33.1, 25.3, 20.7; HRMS (TOF MS ES^+^) calcd for C_24_H_25_N_2_O_5_ [M + H]^+^ 421.1763, found 421.1762. An HCl salt was prepared by dissolving **2b** free base in hot *i*-PrOH (5.0 mL) followed by the addition of concentrated aqueous HCl (0.10 mL, 3 equiv) and cooling to 5 °C. The crystals were filtered and air-dried to give **2b** as its HCl salt. Anal. Calcd for C_24_H_24_N_2_O_5_·HCl·1.5H_2_O (**2b**·HCl·1.5H_2_O): C, 59.64; H, 5.55; N, 5.49; found: C, 59.56; H, 5.83; N, 5.79%. ${\left[\alpha \right]}_{D}^{20}$ −137.9 (*c* 0.6, CHCl_3_/MeOH (1/20), HCl·1.5H_2_O).

**(***7aR,12bS)-9-Hydroxy-3-(4-methylphenethyl)-2,3,4,4a,5,6-hexahydro-1H-4,12-methanobenzofuro[3,2-e]-isoquinolin-7(7aH)-one* (**2c**): The general procedure with **S10** (0.406 g, 1.5 mmol), 1-(2-bromoethyl)-4-methylbenzene (0.502 mL, 3.3 mmol), NaHCO_3_ (0.504 g, 6 mmol), and DMF (10 mL). Purification by SiO_2_ column chromatography afforded **2c** free base (0.330 g, 65%) as a colorless oil. ^1^H-NMR (CD_3_OD): δ 7.11 (d, *J* = 8.4 Hz, 2H), 7.08 (d, *J* = 8.4 Hz, 2H), 6.61 (d, *J* = 8.4 Hz, 1H), 6.56 (d, *J* = 8.4 Hz, 1H), 4.77 (s, 1H), 3.36 (dd, *J* = 5.2 Hz, 2.4 Hz, 1H), 2.97 (d, *J* = 18.4 Hz, 1H), 2.80–2.65 (m, 5H), 2.62 (dt, *J* = 12.8 Hz, 3.2 Hz, 1H), 2.53 (td, *J* = 14.0 Hz, 4.8 Hz, 1H), 2.36 (dd, *J* = 18.4 Hz, 5.6 Hz, 1H), 2.29 (dt, *J* = 13.2 Hz, 3.0 Hz, 1H), 2.28 (s, 3H), 2.21 (td, *J* = 12.0 Hz, 2.1 Hz, 1H), 2.12 (td, *J* = 12.0 Hz, 4.0 Hz, 1H), 1.88–1.82 (m, 1H), 1.68 (d, *J* = 12.4 Hz, 1H), 1.15 (ddd, *J* = 27.0 Hz, 13.4 Hz, 2.4 Hz, 1H); ^13^C-NMR (CD_3_OD): δ 210.3, 144.1, 139.2, 136.9, 135.2, 128.6, 128.2, 127.0, 124.8, 119.6, 117.3, 91.0, 57.0, 56.7, 47.2, 45.1, 41.4, 39.4, 34.6, 33.2, 25.4, 20.2, 19.7; HRMS (TOF MS ES^+^) calcd for C_25_H_28_NO_3_ [M + H]^+^ 390.2069, found 390.2071. An HCl salt was prepared by dissolving **2c** free base in hot MeOH (5.0 mL) followed by the addition of concentrated aq HCl (0.10 mL, 3 equiv) and cooling to 5 °C. The crystals were filtered and air-dried to give **2c** as its HCl salt. Anal. Calcd for C_25_H_27_NO_3_·HCl·1.5H_2_O·0.5CH_4_O (**2c**·HCl·1.5H_2_O·0.5CH_4_O): C, 65.56; H, 6.78; N, 2.92; found: C, 65.37; H, 7.09; N, 2.99%. ${\left[\alpha \right]}_{D}^{20}$ − 139.5 (*c* 0.63, MeOH, HCl·1.5H_2_O·0.5CH_4_O).

*(7aR,12bS)-9-Hydroxy-3-(4-methoxyphenethyl)-2,3,4,4a,5,6-hexahydro-1H-4,12-methanobenzofuro[3,2-e]-isoquinolin-7(7aH)-one* (**2d**): The general procedure with **S10** (0.406 g, 1.5 mmol), 1-(2-bromoethyl)-4-methoxybenzene (0.520 mL, 3.3 mmol), NaHCO_3_ (0.504 g, 6 mmol), and DMF (10 mL). Column chromatography afforded **2d** free base (0.370 g, 64%) as a colorless oil. ^1^H-NMR (DMSO-d_6_): δ 9.09 (s, 1H), 7.13 (d, *J* = 8.4 Hz, 2H), 6.81 (d, *J* = 8.4 Hz, 2H), 6.51 (d, *J* = 8.0 Hz, 1H), 6.46 (d, *J* = 8.4 Hz, 1H), 4.79 (s, 1H), 3.69 (s, 3H), 3.20 (s, 1H), 2.80 (d, *J* = 18.4 Hz, 1H), 2.63–2.53 (m, 5H), 2.51–2.44 (m, 2H), 2.23 (dd, *J* = 18.4 Hz, 5.6 Hz, 1H), 2.12 (d, *J* = 13.6 Hz, 1H), 2.01–1.97 (m, 2H), 1.76–1.72 (m, 1H), 1.47 (d, *J* = 9.2 Hz, 1H), 1.02–0.91 (m, 1H); ^13^C-NMR (DMSO-d_6_): δ 209.3, 157.9, 144.3, 139.6, 132.8, 130.0, 128.0, 124.9, 119.6, 117.3, 114.0, 90.8, 57.2, 57.1, 55.4, 47.1, 44.9, 41.8, 40.1, 35.3, 33.4, 25.5, 21.0; HRMS (TOF MS ES^+^) calcd for C_25_H_28_NO_4_ [M + H]^+^ 406.2018, found 406.2014. An HCl salt was prepared by dissolving **2d** free base in hot i-PrOH (5.0 mL) followed by the addition of concentrated aqueous HCl (0.10 mL, 3 equiv) and cooling to 5 °C. The crystals were filtered and air-dried to give **2d** as its HCl salt. Anal. Calcd for C_25_H_27_NO_4_·HCl·0.5H_2_O·0.5C_3_H_8_O (**2d**·HCl·0.5H_2_O·0.5C_3_H_8_O): C, 66.19; H, 6.83; N, 2.77; found: C, 66.17; H, 6.92; N, 2.91%. ${\left[\alpha \right]}_{D}^{20}$ −136.5 (c 0.68, MeOH, HCl·0.5H_2_O·0.5i-PrOH).

*(7aR,12bS)-9-Hydroxy-3-(4-fluorophenethyl)-2,3,4,4a,5,6-hexahydro-1H-4,12-methanobenzofuro[3,2-e]-isoquinolin-7(7aH)-one* (**2e**): The general procedure with **S10** (0.406 g, 1.5 mmol), 1-(2-bromoethyl)-4-fluorobenzene (0.460 mL, 3.3 mmol), NaHCO_3_ (0.504 g, 6 mmol), and DMF (10 mL). Purification by column chromatography afforded **2e** free base (0.290 g, 50%) as a colorless oil. ^1^H-NMR (CD_3_OD): δ 7.24 (dd, *J* = 8.2 Hz, 5.4 Hz, 2H), 6.99 (t, *J* = 8.8 Hz, 2H), 6.60 (d, *J* = 8.0 Hz, 1H), 6.56 (d, *J* = 8.4 Hz, 1H), 4.77 (s, 1H), 3.30 (s, 1H), 2.97 (d, *J* = 18.4 Hz, 1H), 2.83–2.66 (m, 5H), 2.61 (dt, *J* = 12.8 Hz, 3.4 Hz, 1H), 2.52 (td, *J* = 14.2 Hz, 4.6 Hz, 1H), 2.37 (dd, *J* = 18.6 Hz, 5.8 Hz, 1H), 2.29 (dt, *J* = 14.0 Hz, 2.8 Hz, 1H), 2.22 (td, *J* = 12.0 Hz, 3.0 Hz, 1H), 2.12 (td, *J* = 12.0 Hz, 4.4 Hz, 1H), 1.88–1.82 (m, 1H), 1.69 (d, *J* = 12.4 Hz, 1H), 1.15 (ddd, *J* = 27.2 Hz, 13.2 Hz, 2.4 Hz, 1H); ^13^C-NMR (CD_3_OD): *δ* 210.3, 162.7, 160.2, 144.1, 139.2, 136.0(2), 130.0(2), 127.0, 124.9, 119.6, 117.3, 114.6, 114.4, 91.0, 57.2, 56.6, 47.2, 45.0, 41.5, 39.4, 34.7, 32.8, 25.4, 20.4; HRMS (TOF MS ES^+^) calcd for C_24_H_25_FNO_3_ (M + H^+^) 394.1818, found 394.1825. An HCl salt was prepared by dissolving **2e** free base in hot *i*-PrOH (5.0 mL) followed by the addition of concentrated aq HCl (0.10 mL, 3 equiv) and cooling to 5 °C. The crystals were filtered and air-dried to give **2e** as its HCl salt. Anal. Calcd for C_24_H_24_FNO_3_·HCl·1.5H_2_O (**2e**·HCl·1.5H_2_O): C, 63.14; H, 6.03; N, 2.98; found: C, 63.09; H, 6.18; N, 3.07%. ${\left[\alpha \right]}_{D}^{20}$ − 137.8 (*c* 0.65, MeOH, HCl·1.5H_2_O).

*(7aR,12bS)-9-Hydroxy-3-(3-fluorophenethyl)-2,3,4,4a,5,6-hexahydro-1H-4,12-methanobenzofuro[3,2-e]-isoquinolin-7(7aH)-one* (**2f**): The general procedure with **10** (0.406 g, 1.5 mmol), 1-(2-bromoethyl)-3-fluorobenzene (0.460 mL, 3.3 mmol), NaHCO_3_ (0.504 g, 6 mmol), and DMF (10 mL). Purification by column chromatography afforded **2f** free base (0.270 g, 46%) as colorless oil. ^1^H-NMR (CD_3_OD): *δ* 7.27 (dd, *J* = 14.0 Hz, 8.0 Hz, 1H), 7.05 (d, *J* = 7.6 Hz, 1H), 6.99 (d, *J* = 10.4 Hz, 1H), 6.90 (td, *J* = 8.6 Hz, 2.4 Hz, 1H), 6.61 (d, *J* = 8.4 Hz, 1H), 6.56 (d, *J* = 8.0 Hz, 1H), 4.76 (s, 1H), 3.34–3.32 (m, 1H), 2.96 (d, *J* = 18.4 Hz, 1H), 2.85–2.68 (m, 5H), 2.60 (dt, *J* = 12.6 Hz, 3.4 Hz, 1H), 2.52 (td, *J* = 14.0 Hz, 4.6 Hz, 1H), 2.36 (dd, *J* = 18.4 Hz, 5.6 Hz, 1H), 2.28 (dt, *J* = 13.8 Hz, 2.1 Hz, 1H), 2.21 (td, *J* = 12.4 Hz, 3.2 Hz, 1H), 2.11 (td, *J* = 12.2 Hz, 4.4 Hz, 1H), 1.87–1.81 (m, 1H), 1.67 (d, *J* = 12.4 Hz, 1H), 1.14 (ddd, *J* = 26.8 Hz, 13.4 Hz, 2.4 Hz, 1H); ^13^C-NMR (CD_3_OD): *δ* 210.4, 164.1, 161.7, 144.1, 143.1, 143.0, 139.2, 129.7, 129.6, 127.0, 124.9, 124.3, 124.2, 119.7, 117.4, 115.1, 114.9, 112.5, 112.2, 91.0, 57.2, 56.1, 47.2, 44.9, 41.5, 39.4, 34.7, 33.3, 25.3, 20.4; HRMS (TOF MS ES^+^) calcd for C_24_H_25_FNO_3_ [M + H]^+^ 394.1818, found 394.1821. An HCl salt was prepared by dissolving **2f** free base in hot *i*-PrOH (5.0 mL) followed by the addition of concentrated aqueous HCl (0.10 mL, 3 equiv) and cooling to 5 °C. The crystals were filtered and air-dried to give **2f** as its HCl salt. Anal. Calcd for C_24_H_24_FNO_3_·HCl·1.25H_2_O (**2f**·HCl·1.25H_2_O): C, 63.71; H, 6.13; N, 3.10; found: C, 63.71; H, 6.11; N, 3.11%. ${\left[\alpha \right]}_{D}^{20}$ −149.2 (*c* 0.48, MeOH, HCl·1.25H_2_O).

*(7aR,12bS)-9-Hydroxy-3-(2-fluorophenethyl)-2,3,4,4a,5,6-hexahydro-1H-4,12-methanobenzofuro[3,2-e]-isoquinolin-7(7aH)-one* (**2g**): The general procedure with **S10** (0.406 g, 1.5 mmol), 1-(2-bromoethyl)-2-fluorobenzene (0.460 mL, 3.3 mmol), NaHCO_3_ (0.504 g, 6 mmol), and DMF (10 mL). Purification by column chromatography afforded **2g** free base (0.260 g, 45%) as a colorless oil. ^1^H-NMR (CD_3_OD): *δ* 7.31–7.27 (m, 1H), 7.24–7.19 (m, 1H), 7.11–7.01 (m, 2H), 6.61 (d, *J* = 8.4 Hz, 1H), 6.56 (d, *J* = 8.0 Hz, 1H), 4.77 (s, 1H), 3.36 (dd, *J* = 5.2 Hz, 2.8 Hz, 1H), 2.96 (d, *J* = 18.4 Hz, 1H), 2.94–2.66 (m, 5H), 2.61 (dt, *J* = 12.8 Hz, 3.4 Hz, 1H), 2.40 (td, *J* = 14.0 Hz, 4.6 Hz, 1H), 2.38 (dd, *J* = 18.6 Hz, 5.8 Hz, 1H), 2.29 (dt, *J* = 14.0 Hz, 3.0 Hz, 1H), 2.23 (td, *J* = 12.4 Hz, 3.2 Hz, 1H), 2.12 (td, *J* = 12.2 Hz, 4.6 Hz, 1H), 1.89–1.82 (m, 1H), 1.70–1.67 (m, 1H), 1.15 (ddd, *J* = 27.2 Hz, 13.4 Hz, 2.4 Hz, 1H); ^13^C-NMR (CD_3_OD): *δ* 210.4, 162.4, 160.0, 144.0, 139.2, 130.9, 130.8, 127.8, 127.7, 127.0, 126.8, 126.6, 124.9, 123.9(2), 119.6, 117.3, 114.7, 114.5, 91.0, 57.2, 55.0, 47.2, 45.0, 41.5, 39.4, 34.7, 26.9(2), 25.4, 20.4; HRMS (TOF MS ES^+^) calcd for C_24_H_25_FNO_3_ [M + H]^+^ 394.1818, found 394.1822. An HCl salt was prepared by dissolving **2g** free base in hot *i*-PrOH (5.0 mL) followed by the addition of concentrated aq HCl (0.10 mL, 3 equiv) and cooling to 5 °C. The crystals were filtered and air-dried to give **2g** as its HCl salt. Anal. Calcd for C_24_H_24_FNO_3_·HCl·1.5H_2_O (**2g**·HCl·1.5H_2_O): C, 63.51; H, 5.74; N, 3.05; found: C, 63.09; H, 6.18; N, 3.07%. ${\left[\alpha \right]}_{D}^{20}$ −135.7 (*c* 1.25, MeOH, HCl·1.5H_2_O).

*(7aR,12bS)-9-Hydroxy-3-(4-(trifluoromethyl)phenethyl)-2,3,4,4a,5,6-hexahydro-1H-4,12-methanobenzofuro- [3,2-e]isoquinolin-7(7aH)-one* (**2h**): The general procedure with **S10** (0.406 g, 1.5 mmol), 1-(2-bromoethyl)-4-(trifluoromethyl)benzene (0.554 mL, 3.3 mmol), NaHCO_3_ (0.504 g, 6 mmol), and DMF (10 mL). Purification by column chromatography afforded **2h** free base (0.145 g, 25%) as a colorless oil. ^1^H-NMR (CD_3_OD): δ 7.58 (d, *J* = 8.0 Hz, 2H), 7.45 (d, *J* = 8.0 Hz, 2H), 6.61 (d, *J* = 8.4 Hz, 1H), 6.56 (d, *J* = 8.4 Hz, 1H), 4.77 (s, 1H), 3.34–3.32 (m, 1H), 2.97 (d, *J* = 18.4 Hz, 1H), 2.93–2.72 (m, 5H), 2.60 (dt, *J* = 12.8 Hz, 3.4 Hz, 1H), 2.54 (td, *J* = 14.0 Hz, 4.8 Hz, 1H), 2.38 (dd, *J* = 18.6 Hz, 5.8 Hz, 1H), 2.29 (dt, *J* = 14.0 Hz, 3.0 Hz, 1H), 2.23 (td, *J* = 12.2 Hz, 3.4 Hz, 1H), 2.12 (td, *J* = 12.2 Hz, 4.2 Hz, 1H), 1.88–1.82 (m, 1H), 1.69 (d, *J* = 11.6 Hz, 1H), 1.15 (ddd, *J* = 27.2 Hz, 13.2 Hz, 2.4 Hz, 1H); ^13^C-NMR (CD_3_OD): δ 210.4, 144.1, 139.2, 129.0, 128.5, 128.1, 127.8, 127.5, 127.0, 124.9, 124.8(2), 124.7(2), 119.6, 117.3, 91.0, 57.3, 56.0, 47.2, 44.8, 41.6, 39.4, 34.7, 33.4, 25.3, 20.5; HRMS (TOF MS ES^+^) calcd for C_25_H_25_F_3_NO_3_ [M + H]^+^ 444.1787, found 444.1788. An HCl salt was prepared by dissolving **2h** free base in hot i-PrOH (5.0 mL) followed by the addition of concentrated aq HCl (0.10 mL, 3 equiv) and cooling to 5 °C. The crystals were filtered and air-dried to give **2h** as its HCl salt. Anal. Calcd for C_25_H_24_F_3_NO_3_·HCl·0.5H_2_O·0.5C_3_H_8_O (**2h**·HCl·0.5H_2_O·0.5C_3_H_8_O): C, 61.31; H, 5.73; N, 2.60; found: C, 61.33; H, 5.83; N, 2.70%. ${\left[\alpha \right]}_{D}^{20}$ − 122.0 (c 0.4, MeOH, HCl·0.5H_2_O·0.5i-PrOH).

*(7aR,12bS)-3-(4-Chlorophenethyl)-9-hydroxy-2,3,4,4a,5,6-hexahydro-1H-4,12-methanobenzofuro[3,2-e]-isoquinolin-7(7aH)-one* (**2i**): See the General Procedure in 3.2.

*(4R,7aR,12bS)-3-(4-Bromophenethyl)-9-hydroxy-2,3,4,4a,5,6-hexahydro-1H-4,12-methanobenzofuro[3,2-e]-isoquinolin-7(7aH)-one oxalate* (**2j**): The general procedure with **S10** (0.3 g, 1.11 mmol), 4-bromophenethyl bromide (0.586 g, 2.22 mmol), NaHCO_3_ (0.50 g, 5.95 mmol), and DMF (10 mL) at room temperature. The reaction was heated to 60 °C for 20 h cooled to room temperature, and filtered through a pad of Celite. The DMF was removed via azeotrope with toluene (3 × 20 mL), then purification by SiO_2_ column chromatography with 10% NH_4_OH in MeOH/CHCl_3_ (0% → 5% of 10% NH_4_OH) gave the base **2j** ree base (0.212 g isolated, 41% yield) as a light brown oil. An oxalate salt was prepared by dissolving **2j** in a minimal amount of hot i-PrOH followed by addition of a concentrated solution of oxalic acid in isopropanol. Cooling at 5 °C overnight gave **2j·oxalate** as a precipitate (0.131 g, 44/%), mp 201−204 °C. ^1^H-NMR (CD_3_OD): δ 7.39 (d, *J* = 8.1 Hz, 2H), 7.07 (d, *J* = 8.1 Hz, 2H), 6.71 (d, *J* = 8.1 Hz, 1H), 6.58 (d, *J* = 8.1 Hz, 1H), 4.65 (s, 1H), 3.34 (d, *J* = 0.4 Hz, 1H), 2.96 (t, *J* = 15.8 Hz, 1H), 2.78–2.69 (m, 5H), 2.65–2.61 (m, 1H), 2.38–2.32 (m, 3H), 2.27–2.21 (m, 1H), 2.16–2.07 (m, 1H), 1.84–1.76 (m, 2H), 1.27–1.18 (m, 2H).^13^C-NMR (CD_3_OD): δ 209.12, 144.04, 139.08, 138.89, 131.39, 130.47, 126.78, 125.05, 120.26, 119.89, 118.00, 91.35, 57.67, 56.64, 47.40, 45.03, 42.11, 40.13, 35.16, 33.63, 25.41, 20.98; HRMS (TOF MS ES^+^) C_24_H_24_BrNO_3_ (M+H^+^) 454.1018, found 454.1021. Anal. Calcd for C_24_H_24_BrNO_3_·C_2_H_2_O_4_·2H_2_O (**2j**·C_2_H_2_O_4_·2H_2_O) C, 54.31%; H, 5.15%; N, 2.44%; found: C, 54.09%; H, 5.04%; N, 2.71%. ${\left[\alpha \right]}_{D}^{20}$ −87.0 (c 1.2, MeOH, C_24_H_24_BrNO_3_·C_2_H_2_O_4_·2H_2_O).

*(7aR,12bS)-3-(2,6-Dichlorophenethyl)-9-hydroxy-2,3,4,4a,5,6-hexahydro-1H-4,12-methanobenzofuro[3,2-e]isoquinolin-7(7aH)-one* (**2k**): The general procedure with **S10** (0.56 g, 2 mmol), 2,6-dichlorophenethyl bromide (1.0 g, 4 mmol), NaHCO_3_ (0.70 g, 8.3 mmol), and DMF (10 mL). Purification by column chromatography afforded **2k** free base (0.012 g, 0.02 mmol, 2%) as a yellow oil. The oil was taken up in acetone (2 mL) and oxalic acid (0.002 g) added to give pure **2k·oxalate** (0.014 g) as a white powder. ^1^H-NMR (DMSO-d_6_): δ 7.49 (d, *J* = 8.0 Hz, 2H), 7.32 (t, *J* = 8.0 Hz, 1H), 6.58 (q, *J* = 7.5 Hz, 2H), 4.91 (s, 1H), 3.72–3.70 (m, 1H), 3.24–3.13 (m, 2H), 3.10–2.93 (m, 3H), 2.91–2.81 (m, 1H), 2.76–2.66 (m, 1H), 2.63–2.54 (m, 2H), 2.39–2.32 (m, 1H), 2.20–2.16 (m, 2H), 1.86–1.83 (m, 1H), 1.65–1.62 (m, 1H), 1.06–0.95 (m, 1H); ^13^C-NMR (DMSO-d_6_): δ 208.6, 163.2, 144.4, 140.0, 135.2, 129.9, 129.0, 120.0, 117.7, 90.4, 57.9, 46.2, 31.1, 25.0; m.p. 204-206 °C (decomp); HRMS (ESI): Calcd for C_26_H_26_Cl_2_NO_7_ [M + H]^+^ 444.1133, found: 444.1129; Anal. Calcd for C_26_H_25_Cl_2_NO_7_ ·2.25 H_2_O:C, 54.32; H, 5.17; N, 2.44; found: C, 54.22; H, 4.86; N, 2.59.

### 3.3. In Vitro Pharmacology.

#### 3.3.1. Opioid Receptor Binding Affinity

Frozen whole rat brains excluding cerebellum were thawed on ice, homogenized in 50 mM Tris HCl, pH 7.5 using a Polytron (Brinkman Instruments, Westbury, NY, USA), setting 6 for 20 s), and centrifuged at 30,000× *g* for 10 min at 4 °C. The supernatant was discarded and the pellet was re-suspended in fresh buffer and spun at 30,000× *g* for 10 min. The supernatant was discarded and the pellet was re-suspended to give 100 mg/mL original wet weight. Ligand binding experiments were conducted in polypropylene assay tubes containing 0.5 mL Tris HCl buffer for 60 min at room temperature. [^3^H]DADLE (final concentration 1 nM, PolyPeptide Laboratories, San Diego, CA, USA), [^3^H]DAMGO (final concentration 1 nM, PolyPeptide Laboratories, San Diego, CA, USA) or [^3^H]U69,593 (final concentration 1 nM, Perkin Elmer Life Sciences, Waltham, MA, USA) were used to determine binding at δ-, μ- and κ-opioid receptor sites, respectively. Unlabeled DAMGO (final concentration, 30 nM) was added to the delta assay tubes to block μ-receptor binding. All assay tubes contained 100 μL homogenate suspension. Nonspecific binding was determined in all assays using 0.01 mM naloxone. Incubations were terminated by rapid filtration through Whatman GF/B filters, presoaked in 0.1% polyethyleneimine, using a Brandel R48 filtering manifold (Brandel Instruments Gaithersburg, Maryland). The filters were washed twice with 5 mL cold buffer and transferred to scintillation vials. Cytoscint (MP BioMedicals, Santa Ana, CA, USA) 3.0 mL) was added and the vials were counted the next day using a Perkin Elmer TriCarb liquid scintillation counter. Data were analyzed with GraphPad Prism software (GraphPad Inc., San Diego, CA, USA).

#### 3.3.2. Stimulation of [^35^S]GTPγS Binding

All tissue culture reagents were purchased from Gibco Life Sciences (Grand Island, NY, USA). C6-rat glioma cells stably transfected with a rat MOR or rat DOR [38] and Chinese hamster ovary (CHO) cells stably expressing a human KOR [39] were used for all in vitro assays. Cells were grown to confluence at 37 °C in 5% CO_2_ in Dulbecco’s Modified Eagle Medium (DMEM) containing 10% fetal bovine serum and 5% penicillin/ streptomycin. Membranes were prepared by washing confluent cells three times with ice-cold phosphate buffered saline (0.9% NaCl, 0.61 mM Na_2_HPO_4_, 0.38 mM KH_2_PO_4_, pH 7.4). Cells were detached from the plates by incubation in warm harvesting buffer (20 mM HEPES, 150 mM NaCl, 0.68 mM EDTA, pH 7.4) and pelleted by centrifugation at 1600 rpm for 3 min. The cell pellet was suspended in ice-cold 50 mM Tris-HCl buffer, pH 7.4, and homogenized with a Tissue Tearor (Biospec Products, Inc., Bartlesville, OK, USA) for 20 s. The homogenate was centrifuged at 15,000 rpm for 20 min at 4 °C. The pellet was re-homogenized in 50 mM Tris-HCl with a Tissue Tearor for 10 s, followed by re-centrifugation. The final pellet was re-suspended in 50 mM Tris-HCl and frozen in aliquots at 80 °C. Protein concentration was determined via a BCA protein assay (Thermo Scientific Pierce, Waltham, MA, USA) using bovine serum albumin as the standard.

Agonist stimulation of [^35^S]guanosine 5′-O-[γ-thio]triphosphate ([^35^S]GTPγS, 1250 Ci, 46.2 TBq/mmol) binding to G-protein was measured as described previously [40]. Briefly, membranes (10−20 μg of protein/tube) were incubated 1 h at 25 °C in GTPγS buffer (50 mM Tris-HCl, 100 mM NaCl, 5 mM MgCl2, pH 7.4) containing 0.1 nM [35S]GTPγS, 30 μM guanosine diphosphate (GDP), and varying concentrations of test compound. G-Protein activation following receptor stimulation of [^35^S]GTPγS (% stimulation) with test compound was compared with 10 μM of the standard compounds [D-Ala2,N-MePhe4,Gly-ol]- enkephalin (DAMGO) at MOR, D-Pen2,5-enkephalin (DPDPE) at DOR, or U69,593 at KOR. The reaction was terminated by vacuum filtration of GF/C filters that were washed 10 times with GTPγS buffer. Bound radioactivity on dried filters was determined by liquid scintillation counting, after saturation with EcoLume liquid scintillation cocktail, in a Wallac 1450 MicroBeta (PerkinElmer, Waltham, MA, USA). The results are presented as the mean ± standard error (SEM) from at least three separate assays performed in duplicate; potency (EC50 (nM)) and % stimulation were determined using nonlinear regression analysis with GraphPad Prism.

#### 3.3.3. Forskolin-Induced cAMP Accumulation Assays

##### Cell Lines and Cell Culture

HitHunter Chinese hamster ovary cells (CHO-K1) that express human μ-opioid receptor (OPRM1), human κ-opioid receptor (OPRMK1), and human δ-receptor (OPRMD1) were used for the forskolin-induced cAMP accumulation assay. PathHunter CHO cells expressing human μ-opioid receptor β-arrestin-2 EFC cells were used for the β-arrestin-2 EFC recruitment assay. Both cell lines were purchased from Eurofins DiscoverX (Fremont, CA, USA). Cell culture was performed as previously described [41].

##### Assays

These were performed as previously described using HitHunter CHO-K1 cells expressing either human OPRM1, OPRK1, or OPRMD1 cells. Briefly, cells were dissociated from culture plates and plated at 10,000 cells/well in a 384-well tissue culture plate and incubated overnight at 37 °C in 5% CO_2_. Stock solutions of compound were made in 100% DMSO at a 5 mM concentration. A serial dilution of 10 concentrations was made using 100% DMSO, creating 100× solutions of the compound for treatment. The 100× solutions were then diluted to 5× solutions using assay buffer consisting of Hank’s Buffered Salt Solution, HEPES, and forskolin. The HitHunter cAMP Assay for Small Molecules by DiscoverX was then used according to manufacturer’s directions, utilizing the 5× solutions containing the compound studied. Cells were incubated with compound for 30 min at a 1× concentration. The following day, the Cytation 5 plate reader and Gen5 Software were used to quantify luminescence (BioTek, Winooski, VT, USA) [41].

#### 3.3.4. β-Arrestin-2 EFC Recruitment Assay

Assays were performed as previously described [41] using PathHunter human μ-opioid receptor β-arrestin-2 EFC cells. Briefly, cells were dissociated from culture plates and plated at 5000 cells/well in a 384-well tissue culture plate and incubated overnight at 37 °C in 5% CO_2_. Stock solutions of compound were made in 100% DMSO for a final concentration of 5 mM. A serial dilution of 11 concentrations was made using 100% DMSO to create 100× solutions of the compound. Assay buffer containing Hank’s Buffered Salt Solution and HEPES was used to dilute the 100× solutions to 5× solutions. The DiscoverX PathHunter assay was used according to manufacturer’s instructions. Cells were treated with the compounds for a final 1× concentration for 90 min. at 37 °C and 5% CO_2_. Reagents from the assay kit were used accordingly and the cell culture plate was protected from light for 1 h. Cytation 5 plate reader and Gen5 Software were used to quantify luminescence (BioTek).

#### 3.3.5. Data Analysis

Data were analyzed as previously described [42] using GraphPad Prism 6.0 software (GraphPad, San Diego, CA, USA different address given before). Briefly, sigmoidal dose-response curves in the forskolin-induced cAMP accumulation assay and the β-arrestin2 EFC recruitment assay were generated using nonlinear regression analysis. Compounds were evaluated in triplicate in individual experiments with *n* ≥ 2. All values in the cAMP accumulation assay and β-arrestin2 recruitment assay are reported as the mean ± SEM. Bias factors were calculated using Equation (1) shown below.
(1)$$\begin{array}{ccc} Log\;\left(bias\;factor\right) & = & {\left(Log\;\left(\frac{EC50\;test\times Emax\;DAMGO}{Emax\;test\times EC50\;DAMGO}\right)\right)}_{cAMP\;}\\ & - & {\left(Log\;\left(\frac{EC50\;test\times Emax\;DAMGO}{Emax\;test\times EC50\;DAMGO}\right)\right)}_{\beta -arrestin} \end{array}$$

### 3.4. In Vivo Pharmacolog

#### 3.4.1. Measurement of Respiration Rate and Arterial Oxygen Saturation in Mice

Male Swiss Webster mice (Taconic Biosciences, Germantown, NY, USA) weighing 30–35 g were used. Mice were housed in a temperature- and humidity-controlled environment with a 12-h light-dark cycle in the Temple University Animal Care Facility. They were supplied with food and water ad libitum. Before any procedure was applied, the mice were acclimated for 1 week in the animal facility. Behavioral testing was performed between 11:00 a.m. and 5:00 p.m. On the day of the experiment, mice were brought to the room and acclimated for 45–60 min in the observation boxes All animal care and experimental procedures were approved by the Institutional Animal Care and Use Committee of Temple University (Protocol #4793 Respiration Measurement in Mice; Approval Date: 1 June 2018), and conducted according to the NIH Guide for the Care and Use of Laboratory Animals.

Respiration and oxygen saturation (SpO_2_) were measured using MouseOx Plus Rat and Mouse Pulse Oximeter (Starr Life Sciences Corp, Oakmont, PA, USA) in conscious, freely moving animals. Animals were exposed to 4% isoflurane for 30 s to connect throat collar sensor and to inject (s.c.) either saline, morphine 10 mg/kg, or **2i** (0.01–0.1 mg/kg, *n* = 6–8). Mice were then placed into observation boxes and recording was started 5 min later to eliminate any anesthesia effect. Respiration and SpO_2_ were recorded every second and averaged over 1-min periods for 40 min. Morphine, 10 mg/kg, was used as a positive control [43].

#### 3.4.2. Statistical Analysis

Area under the curve (AUC) was calculated from 6 min to 45 min and analyzed using one-way analysis of variance (ANOVA) followed by Dunnett’s multiple comparison test. Data are expressed as mean ± standard error of the mean (S.E.M.), and *p* < 0.05 was accepted as statistically significant. GraphPad Prism, version 7, was used for data analysis.

### 3.5. Warm-Water Squirrel Tail-Withdrawal and Operant Responding

Male squirrel monkeys (Saimiri sciureus) were housed in a climate-controlled vivarium with a 12-h light/dark cycle (7 AM–7 PM) in the McLean Hospital Animal Care Facility (licensed by the U.S. Department of Agriculture and compliant with guidelines provided by the Committee on Care and Use of Laboratory Animals of the Institute of Laboratory Animals Resources, Commission on Life Sciences, National Research Council; 2011). Tail withdrawal latencies and food-maintained behavior were assessed as described previously [44]. Briefly, monkeys were seated in customized Plexiglas chairs that allowed their tails to hang freely behind the chair and were equipped with colored stimulus lights, a response lever, and a receptacle into which 0.15 mL of 30% sweetened condensed milk could be delivered. Animals were trained to respond under a fixed-ratio 10-response (FR10) schedule of food reinforcement in the presence of red stimulus lights. Completion of 10 responses in less than 20 s resulted in milk delivery, and initiated a timeout (TO) period of 30 s during which all stimulus lights remained off. Failure to complete 10 responses within 20 s initiated the 30 s TO. Tail withdrawal latencies were measured during the 30 s TO periods by immersing the subject’s tail in water held at 35 °C, 50 °C, 52 °C, or 55 °C (each temperature of water was presented in a randomized order). Experimental sessions were 4 or 5 sequential cycles, each composed of a 10 min TO during which no lights were on and responding had no programmed consequences followed by a 5 min response component during which the FR10 schedule of food reinforcement and interspersed determinations of tail withdrawal latencies was in effect. Cumulative doses of **2i** or morphine were administered shortly after the onset of the 10 min TO.

### 3.6. Squirrel Monkey Ventilation

Male squirrel monkeys were acclimated to a customized round acrylic chamber (13.75″ day × 15″ h) that served as a whole-body plethysmograph (EMKA Technologies, Montreal, PQ, Canada). Gas (either air or a 5% CO_2_ in air mixture) was introduced to and extracted from the chamber at a constant flow rate of 5 L/min. Experimental sessions consisted of 4–6 consecutive 30 min cycles, each comprising a 15 min exposure to air followed by a 15 min exposure to 5% CO_2_. Drug effects were determined using cumulative dosing procedures, and injections were administered following each exposure to 5% CO_2._ Respiratory rate and tidal volume (mL/breath) were recorded over 1 min periods and were multiplied to provide minute volumes. Data from the last three minutes of each exposure to air or CO2 were averaged and used for analysis of drug effects on ventilation.

## 4. Computational Methods

All the ligands considered in this study and their conformers (see Appendix A) in their protonated form were geometry optimized via quantum chemical (QM) calculations at the B3LYP/6-31G* level in the gaseous phase as implemented in Gaussian 09 software [45]. The atomic polar tensor derived charges from these calculations were used to assign a partial charge on each atom for the ligands. All other parameters were determined by chemical analogy with the topology and the parameters files of the all-atom CHARMM force field (version c42b2) [46]. The structures of the inactive forms of MOR (PDB # 4DKL) and DOR (4EJ4) were taken as the starting configurations. After removing the cocrystallized ligands and crystal water/ions, the intracellular loop (ICL3) connecting TM5 and TM6 was first modeled in each receptor using MM (ab initio) methods [8,47,48]. The modeled receptors were then embedded in a membrane composed of zwitterionic 1-palmitoyl-2-oleoyl-phosphatidylcholine lipid molecules with initial concentrations of Na^+^, K^+^, and Cl^–^ ions in the extracellular (EC) and intracellular (IC) regions. All the amino acids were assumed to be neutral at physiological pH except Asp^–^, Glu^–^, Lys^+^ and Arg^+^; additional ions were used to neutralize the systems. The initial orientation and relative position of the receptor with respect to the membrane were obtained from the OPM database [49]. The receptor and the membrane were then solvated in TIP3P water, and a single Na^+^ ion was introduced in the binding pocket and coordinated as in the DOR (4N6H) [50]. The system was then energy minimized and thermally equilibrated according to the following protocol: first, the conformation of the receptor and Na^+^ were kept fixed, and the membrane and water were gradually heated to the target temperature (37 °C) at constant pressure (1 atm); the system was then equilibrated for 5 ns; the constraints on the side chains and ion were then removed, and the system equilibrated for another 5 ns; finally, all the constraints were removed, and the system equilibrated for another 5 ns. Throughout the entire equilibration process, the ion remained coordinated with the anchoring aspartic acid D147 (MOR) and D128 (DOR) (sequence numbering follows the corresponding crystal structures). A conformation (snapshot) of each system at the end of the final equilibration phase was used to dock the ligands. These are the basal conformations that all the ligands “see” before entering the binding pocket, and were used for all the comparative analyses, regardless of the experimentally determined activity or pharmacological outcome. After removing the Na^+^ ion, the ligands (and their conformers) were rigidly docked into the binding pocket based on two criteria: the close contact between the charged amine and the anchoring Asp^–^ and the binding mode of the antagonists β-FNA and naltrindole co-crystallized with the MOR and the DOR, respectively. Several conformers (cf. Appendix A) could be eliminated by steric considerations alone. Others (especially those involving rotations of ϕ_1_; cf. Appendix A) could still dock without apparent steric clashes upon small relaxations of the pocket side chains, but they tended to lose the critical interaction with Asp^–^ in the course of equilibration or early production; when this occurred in repeated simulations, the conformer was discarded. Overall, between one and four stable conformers were left for each ligand (see Appendix A). The three-stage equilibration protocol described above for Na^+^ was repeated for each ligand/conformer after docking. Steric relaxation of the ligand and residues in the pocket set in during the early stages of equilibration. Five independent 50 ns molecular dynamics simulations were conducted for each system at 37 °C and 1 atm, using periodic boundary conditions and particle mesh Ewald summations. This simulation time was sufficient to ensure convergence and statistical analysis of the quantities of interest (hydrophobic contacts, H-bonds, electrostatic interactions), which were computed after structural relaxation set in (estimated from Cα-RMSD vs. time), typically during the last half of the dynamic trajectory (production run). The results combine data from all the independent simulations for each ligand conformer/OR.

## 5. Conclusions

We hypothesized that substituents at the tail end of the body of a large molecule might modify the in vitro activity and/or potency of the compound and possibly modify a G-protein biased compound that acted primarily through MOR to a bifunctional ligand. We found that a substituent, a chlorine atom, modified the activity of *N*-phenethylnorhydromorphone (**S11**), a potent full agonist with a DOR-MOR δ/μ potency ratio of 38.5, to a compound with a δ/μ potency ratio of 1.2, *N*-*p*-chlorophenethylnorhydromorphone (**2i**). It exhibited potent partial MOR agonist and potent full DOR agonist activity. In fact, the introduction of a *p*-Cl substituent (**2i**) in the *N*-phenethyl moiety did not particularly reduce the MOR potency of **S11** but instead increased its DOR potency; it induced a change from a molecule that acted primarily as a MOR ligand to a bifunctional compound with the ability to interact potently with MOR and DOR. This change was due to a simple substituent at the tail end of the compound. Molecular modeling and simulations found that the substituent on the aromatic ring of the *N*-phenethyl moiety is located in an area where relatively small changes in the *N*-phenylethyl ring via substitution perturb residues located in quite different regions of the opioid receptors and engage different TMHs. In theory, the combination of MOR and DOR properties found in **2i** might have made the compound less likely than other potent analgesics to cause respiratory depression [17]. Indeed, that was found to be the case in mice using normal air, where a clear difference was found between the effects of **2i** and morphine on respiration. Both **2i** and morphine are partial agonists; if the lack of effect on respiration was due to the partial agonist character of **2i**, the same effect would be expected with morphine. The ability of **2i** to recruit β-arrestin2 at least as well as morphine (Table 2) would predict that it should exhibit all of the side-effects known to occur with morphine. The inability of **2i** to depress respiration in mice might indicate that the recruitment of β-arrestin2 may not be the cause of all of the side-effects seen with opioids [4]. However, with squirrel monkeys under more stringent conditions, in an assay using 5% CO_2_ mixed in air, **2i** was found to be as effective as morphine in depressing respiration. Further work is necessary to determine whether **2i** will produce the gastrointestinal effects, tolerance, and dependence that occur with other G-protein biased opioids.

## 6. Patents

K. C. Rice, A. E. Jacobson, F. Li, E. Gutman, E. W. Bow: Biased Potent Opioid-Like Agonists as Improved Medications to Treat Chronic and Acute Pain and Methods of Using the Same. International Application PCT/US19/22701, 18 March 2019 (PCT Application Serial No.: PCT/US2019/022701, filed 18 March 2019). Patent No. WO 2019182950.

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
