# Peer review of "The Intriguing Effects of Substituents in the N-Phenethyl Moiety of Norhydromorphone: A Bifunctional Opioid from a Set of “Tail Wags Dog” Experiments"

_molecules, 2020, doi:10.3390/molecules25112640_

Round 1
Reviewer 1 Report
By means of N-substituent adjustment on morphine and oxymorphone compound, the binding of these compounds to receptors can be changed, and even cause agonists and antagonists to reverse or partially reverse. This research is of great significance for the application of opiod compounds to avoid some of its side effects. The background, structural design of derivatives and structural characterization of synthetic process optimization in the paper are relatively standard, and the activity evaluation and SAR discussions are basically professional. Compound 2i has some research value. So, I think this article is fit for publication in the journal.Author Response
Reply to Reviewer 1:
Thank you for your review.
Reviewer 2 Report
In this manuscript, Wang et al. examined the impact of various substituents in the N-phenethyl moiety of norhydromorphone. The authors show that some substituents (cyanoalkyl type, non-substituted aromatic rings, phenyl ring substituted with a nitrogen dioxide) displayed antagonistic properties. Substitution of the phenyl ring with fluoride or trifluoromethyl led to agonist activity at mu and delta receptors with high affinity to the mu opioid receptor and a 10-time mu over delta selectivity. Substitution with chloride (compound 2i) increased both affinity and potency to the delta opioid receptor suggesting that this compound could be a promising candidate with dual mu-delta activity that would produce analgesia and reduced side effects. This compound indeed induced no respiratory depression in normal air condition in mice but showed a profile similar to morphine in squirrel monkeys in the presence of 5% CO2. Compound i2 relieved thermal analgesia as efficiently as morphine at 50ºC and 52ºC and partially relieved it at 55ºC. However, behavioural impairment took place at all analgesic concentrations as for morphine. Detailed modelling studies are also provided to explore interactions with critical amino acids within the binding pocket of mu and delta opioid receptors.
Overall, the manuscript is clearly written. Conclusions are supported by the experimental data and in silico analysis provides useful insights for the design of novel ligands.
Here follow a few questions to the authors
- Compound 2b is an antagonist in the GTPgS assay but a full agonist in the cAMP assay. Despite the obvious amplification taking place between the two steps, this is however surprising. How do the authors explain this difference?
- The authors indicate that compound 2i is the “most interesting candidate” among those of group 4 (based on table 2). Compounds 2f and 2g with the fluoride substituents seem very good candidates as well. Could the authors further explain the reasons of their choice?
- If I am correct, the structure used for the docking studies is the inactive form of the receptors. Could the authors explain their choice? How is this more relevant than the structure of the active form of the mu opioid receptor?
- Recently, Drumiscuta et al (reference 33) reported phenethyl modification of oxymorphone, a compound very similar to norhydromorphone. Amino acids identified as contact points in the binding pocket seem different in the two studies. Could the authors discuss this apparent discrepancy?
- DADLE is not selective for the delta receptor. The Kd for the rat delta opioid receptor is about 2nM and that for the rat mu opioid receptor about 4nM. Could the authors justify the choice of this ligand? Could also the authors explain why they preferred the use of agonists rather than antagonists to determine the Ki values of their compounds?
- Is the final concentration of DMSO indeed 1% in the cAMP and beta-arrestin assays? If so, this concentration is quite high and affects the membrane properties which can somehow impact the results.
Author Response
Reply to Reviewer 2.
• Compound 2b is an antagonist in the GTPgS assay but a full agonist in the cAMP assay. Despite theobvious amplification taking place between the two steps, this is however surprising. How do the authors explain this difference?
REPLY: We were quite surprised by these results. This is most likely explained by the amplification of the adenylyl cyclase response compared to the proximal G protein response, difference in the cell lines for the two assays and the fact that one assay used membrane and the other whole cells.
• The authors indicate that compound 2i is the “most interesting candidate” among those of group 4 (based on table 2). Compounds 2f and 2g with the fluoride substituents seem very good candidates as well. Could the authors further explain the reasons of their choice?
REPLY: We agree that the fluoride compounds are interesting MOR compounds, especially the trifluormethyl compound 2h. However, please note that none of them had sufficient potency at the DOR (as compared to their MOR potency). We were seeking bifunctional compounds, MOR-DOR compounds with a low ratio, and we did not find them among the fluorides. For example, in Table 1, if you look at the DOR/MOR potency ratio for 2e (p-fluoro) the ratio was calculated as = 25, that of 2f = 22, 2g = 44; 2h = 11, and 2i = 1.2. Also, I thought that the combination of a low efficacy potent MOR compound (in the GTP assay) and a high efficacy potent DOR compound might prove to be an interesting combination. I was not familiar with many other compounds that had those characteristics. I thought that the MOR low efficacy might lessen side-effects and high DOR potency might increase analgesia in vivo. I had no basis for those suppositions, just thought it was of sufficient interest to pursue it further.
• If I am correct, the structure used for the docking studies is the inactive form of the receptors. Could the authors explain their choice? How is this more relevant than the structure of the active form of the mu opioid receptor?
REPLY: We have discussed our choice in the Methods section. The inactive conformations (PDB: 4DKL and 4EJ4) were used initially, but they were post-processed to obtain standard (basal) references for all comparative analyses. These are the states “seen” by all the ligandswhen first enter a binding site, regardless of their effects on the OR during the dynamics (using an active state would assume the ligands activate the receptor).
• Recently, Drumiscuta et al (reference 33) reported phenethyl modification of oxymorphone, a compound very similar to norhydromorphone. Amino acids identified as contact points in the binding pocket seem different in the two studies. Could the authors discuss this apparent discrepancy?
REPLY: We renumbered the sequences to make it easier to compare. The contacts of the body (Fig. 1) are similar to those found in Ref. 33. The difference observed (H297 and K233) may be attributed to the different structures used in the simulations: we used inactive 4DKL and 4EJ4, whereas Drumiscuta et al. used the active 5C1M. The other contacts seen in [33] are contained in Fig. 2. See Fig. S1 for a summary.
• DADLE is not selective for the delta receptor. The Kd for the rat delta opioid receptor is about 2nM and that for the rat mu opioid receptor about 4nM. Could the authors justify the choice of this ligand?
REPLY: DADLE was made selective. See section 4.3: "Unlabeled DAMGO (final concentration, 30 nM) was added to the delta assay tubes to block mu-receptor binding."
• Could also the authors explain why they preferred the use of agonists rather than antagonists to determine the Ki values of their compounds?
Reply: We are aware that using a radiolabeled agonist to assess binding to G-protein coupled receptors can reveal two apparent affinity states, whereas antagonists do not have this potential. However, under the conditions employed with the well-studied assays that we used,there was no evidence of multiple binding sites.
• Is the final concentration of DMSO indeed 1% in the cAMP and beta-arrestin assays? If so, this concentration is quite high and affects the membrane properties which can somehow impact the results.
Reply: Yes, the final concentration is 1%. We have not seen negative effects on cellular responses in these assays as our control compounds are treated the same way and we have not seen any abnormal responses. Additionally, this concentration is within the recommended limit per the manufacturer's instructions.
Reviewer 3 Report
- The manuscript by Wang et al. reports synthesis, biological evaluation and molecular modelling of 15 novel norhydromorphone derivatives with different N-substituents, in particular, with different decoration patterns at the aromatic ring of the N-phenethyl.
- The syntheses were accomplished by alkylation of the secondary amine (N-norhydromorphone). The compounds were tested for opioid affinities and function at the receptors following typical methods. Further, one derivative was assayed in vivo for antinociception and some typical opioid side-effects.
- The authors put forward some rationalization based on molecular modelling as to why the minor changes in structure produce the observed changes in activity (affinity/function)
- Overall, the account is well-written, the study is designed in a correct manner and the results are very interesting. The effects the substitution patterns 'at the tail' are reported to have on the biological activity are indeed, as the title states, intriguing. Modest modifications like shifting nitro group from para- (2a) to meta- (2b) position of the ring reverse the function of the compound (mOR agonist/antagonist) while only moderately affecting the affinity (at mOR). The compound (2i) is a very interesting one. In general, several compounds have very high affinity/potency, interesting functional properties etc.
- The work deserves publication in Molecules. There are however some technical points that need to be fixed (see point 7 for the detailed list). Thus I recommend the paper be accepted after minor/major revisions.
- Further, I am not satisfied with the way the molecular modelling part is presented (and perhaps also performed).
- DESIGN: Did the authors perform more than one repetition of MD simulation? MD is a method based on statistical mechanics. State-of-the-art standard is to perform several replicas/repetitions of the simulations, and to collect statistics of contacts, dihedrals, rmsds etc. IF THE AUTHORS DID NOT PERFORM at least three production runs for each considered compound, THEY SHOULD SUPPLEMENT THIS. These days it should not take much time. I am afraid that the conclusions taken with the analysis of a single MD run are not necessarily sound.
- METHODS:
- The description of the modelling results should start with telling the reader in one sentence what was the modelling procedure and which compounds were modelled in complex with which receptors. This should be also EXPLICITLY stated in the methods. Which receptors were modelled? Which receptor structures? Was the receptor state active-like or inactivated? Which compounds?
- What happened to the antibody/G-protein/an artificial stabilizing sequence (if present) in crystal structures? Were there any missing loops? If there were, how were they modelled? Please add this information to methods.
- What was the software used for modelling? Please add this to method description.
- How the authors chose the binding pose for molecular dynamics (MD) simulations? Please add this information to methods. Reference to a previous paper is not enough, at least a sentence or two should be given.
- RESULTS’ PRESENTATION (but approach this only after having at least 3 repetitions for each compound!!!) :
- The modelling results part in fact lacks a proper description of results. The reader must guess that the compounds dock to the usual binding site of opioids, with a canonical interaction of charged amine with the aspartic acid. He/she also must guess that the N-substituent extends deeper down towards the intracellular part, entering the subpocket that was rather rarely considered to be opioid binding site (‘deep binding site’). On the other hand, there are some crystals or modelling reports that suggested ‘a deep binding’ before for other receptors. (e.g. 10.1039/C8SC01680A). If I interpret the Figures and the description correctly, the N-substituent goes deeper than ‘usually’. It deserves emphasis. Even more, if we consider that the residues in this ‘deeper site’ are directly involved in the receptor activation (e.g. W6.48 -> please look up: 10.1002/anie.201302244, 10.1002/anie.201501742, 1038/5733, 10.1002/anie.201409679,)
- The authors do not compare the binding modes of the described compounds to the binding poses present in available opioid receptor crystal structures.
- N-phenethyl substituent is present in fentanyl structure. Could the authors compare positioning of N-phenethyl in their structures with that of this substituent in fentanyl? There are some recent models of fentanyl binding to mOR (1007/s00894-019-3999-2, 10.3390/molecules24040740) and some older ones (10.1021/jm9903702)
- The authors claim that they equilibrated the simulations. Please provide the RMSD plots of the receptor helices (backbone of the helices) and of the ligand position (body and tail, in separate). Please add them to SI.
- The authors write that some contacts were frequent etc. Please provide the frequency plots, occupancy of the contacts.
- The authors claim that “iii) Relative affinities correlate with the number and frequency of favorable contacts (polar/nonpolar, H-/halogen-bonding) between the substituent group and the OR; other contacts (body- and phenethyl ring-OR) are common to all the ligands/conformers and thus less critical in modulating the OR behavior.” Is this a quantitative correlation or merely a qualitative “opinion”? Please provide a correlational plot and the statistics of correlation.
- Please analyse the dihedral angles (means +/- S.D. taken from a few repetitions) of binding site residues’s side chains contacting the ligands. Are there any (quantitative) correlations with the experimental affinity/activity? With the substituent volumes? You could look at these examples: e.g. 3390/molecules24040740, here the authors found correlations between W6.48 dihedral angles and the volume of N-substituent in fentanyls. A correlation between W6.48 dihedrals and the functional activity was suggested in 10.1016/j.jmb.2017.05.009. Do the authors have similar relationships in their data? Finding such and other correlations (or lack thereof) would enable the authors to include results for mor than just 2e and 2i.
- The AA numbering in Figure 1 and Figure 2 and S3 is at least partially incorrect. think that in mOR the aspartic acid anchoring the amine is D147 (in murine/rat sequence), W of the TM6 might be W293 etc. etc. Please check this carefully! You can consider using the Ballestros-Weinstein numbering, this would facilitate comparisons between mOR and dOR.
- Figure S1A should have some captions allowing the reader to have some orientation ‘at which depth’ these blue, red, yellow green etc. residues are present. Please mark e.g. D3.32, W6.48 or alike to provide orientation for the reader. Which receptor is shown here? Which structure.
- Minor (‘editorial’) issues.
- The manuscript is divided in a strange manner. There is a section ‘2. Results and Discussion’ and a section “3. Biological Results and Discussion”. This is unusual and as per my reading of MDPI author guidelines it is not allowed. Further, modelling is given as “subsection” of “in vitro studies” which is strange.
- Some parts of the text seem to be placed in the wrong place. For example:
- In the ‘Chemistry’ subsection, the authors describe the preparation of S10, then they talk about what novel compounds they report in this contribution. Then all of a sudden they talk about binding data, or functional assays. That these assays were done and their results are in Tables this and that, should start the subsection that follows. I cannot understand why 2a-2i and 2j/2k are separated in “chemistry”, there seem not anything ‘chemically’ special about them. That they were tested for cAMP accumulation and not for binding/GTPgS, should be explained in the sections that follow. By the way – why was it so?
- The last paragraph of 2.2.2. seems (perhaps) better placed at the beginning 2.2.3.
- Why are the results of the antinociceptive test given under 3.2. respiratory depression studies? Ought there not to be a separate subsection for them?
- P2. (the 2nd paragraph). “Tolerance[1-4]” -> please add space.
- P3. (last paragraph of introduction) – I am not English native speaker (and many of the authors seem to be), but it seems to me that ‘the question of what… has not hitherto been ascertained’ is not correct. I do not think one can ‘ascertain a question’, but I may be wrong, please check this.
- P3. (bottom of the page) I think “secondary amine 10” should be “secondary amine S10”.
- page 4, Scheme 2.
- The compounds should be labelled "1a-d" and "2a-k" so that it is clear that there are more than one as a result of a given reaction series.
- As to the bottom part: it seems that the usage of "R2" is inconsistent. With the reaction arrow, it denotes the alkyl chain to which Br atom is attached. With the structure 2 (bottom, right) and the box with red borders, it denotes a substitution pattern at the N-phenethyl ring. To me, it seems inconsistent. Please correct.
- There are two dots after the Scheme caption.
- P5. The table caption contains some strange characters instead of mu, delta, kappa and gamma. This is perhaps only the issue of the pdf generated by the submission system, but please check this to be sure.
- P6 (last paragraph): Ki = < 1 nM should rather read Ki < 1 nM
- page 17, synthesis
- Why 2i is described before 1a, 1b, 2a… etc? Please consider putting this after 2h and before
- 2i. HCl is described in a separate paragraph while obtaining of other HCl salts is given together with their respective bases. Please make this consistent.
- Please make 2i, 1a, 1b, 1c, 1d, 2a, 2b, 2c, 2d, 2e, 2f, 2g in paragraph headings in bold.
- EC50 is written without subscript. On the other hand ED50 is written with subscript. This should be consistent. I think the subscript is these days more common, so my advice would be to use it here. At least, please, make the usage consistent (there are many instances of EC50 in the text, I shall not enumerate them here). Similarly, I would personally prefer Ki with ‘i' in subscript.
- Page 22. 4.3. “contained100 uL” -> please add space.
- Page 22. Why is 4.3 “Pharmacology. Opioid …” not under ‘In vitro pharmacology” and other assays are?
- SI, NMRs. The NMRs are not numbered. It is a little inconvenient for the reader that they have no textual ‘description’ like ‘2a’, ‘2b’ etc. Please add at least such a label.
- SI, NMRs. The structures on p. 19 and 20 (S11 derivative) have different stereochemistries than S11 in Table S1, Table 2 and all other derivatives in body and SI. Please check this and correct.
- SI, Scheme S1. Please note in the figure caption that the percent values are yields, are they? In the lower part, what was the yield for S10 to S11 conversion? I think “10 ~ 70 %” applies to the diverse derivatives 2a-j discussed in the body, but not to S11. Please correct.
- SI, Figure S2. The Figure caption says that the maps were constructed with ‘B3LYP/DGDZVP’. Were the structures optimized? Please note this.
Author Response
Reply to Reviewer 3.
6. Further, I am not satisfied with the way the molecular modelling part is presented (and perhaps also performed).
a. DESIGN: Did the authors perform more than one repetition of MD simulation? MD is a method based on statistical mechanics. State-of-the-art standard is to perform several replicas/repetitions of the simulations, and to collect statistics of contacts, dihedrals, rmsds etc. IF THE AUTHORS DID NOT PERFORM at least three production runs for each considered compound, THEY SHOULD SUPPLEMENT THIS. These days it should not take much time. I am afraid that the conclusions taken with the analysis of a single MD run are not necessarily sound.
REPLY: We have discussed these points in the Methods. Multiple simulations were needed not only for statistical reasons but also to eliminate unlikely conformers: some conformers tended to lose the critical H-bond between the charged amine and the anchoring Asp- during equilibration or early production (Table S2), in which case a new simulation was conducted. It was discarded if it became unstable in more than two runs.
b. METHODS:
i. The description of the modelling results should start with telling the reader in one sentence what was the modelling procedure and which compounds were modelled in complex with which receptors. This should be also EXPLICITLY stated in the methods. Which receptors were modelled? Which receptor structures? Was the receptor state active-like or inactivated? Which compounds?
REPLY: We have expanded the Methods to make the paper self-contained.
ii. What happened to the antibody/G-protein/an artificial stabilizing sequence (if present) in crystal structures? Were there any missing loops? If there were, how were they modelled? Please add this information to methods.
REPLY: All this is now clarified in the Methods.
iii. What was the software used for modelling? Please add this to method description.
REPLY: It is now clarified. We used CHARMM (for receptor modeling and dynamics simulations) and Gaussian (for quantum chemical calculations).
iv. How the authors chose the binding pose for molecular dynamics (MD) simulations? Please add this information to methods. Reference to a previous paper is not enough, at least a sentence or two should be given.
REPLY: We have now described it in the Methods.
c. RESULTS’ PRESENTATION (but approach this only after having at least 3 repetitions for each compound!!!) :
i. The modelling results part in fact lacks a proper description of results. The reader must guess that the compounds dock to the usual binding site of opioids, with a canonical interaction of charged amine with the aspartic acid. He/she also must guess that the N-substituent extends deeper down towards the intracellular part, entering the subpocket that was rather rarely considered to be opioid binding site (‘deep binding site’). On the other hand, there are some crystals or modelling reports that suggested ‘a deep binding’ before for other receptors. (e.g. 10.1039/C8SC01680A). If I interpret the Figures and the description correctly, the N-substituent goes deeper than ‘usually’. It deserves emphasis. Even more, if we consider that the residues in this ‘deeper site’ are directly involved in the receptor activation (e.g. W6.48 -> please look up: 10.1002/anie.201302244, 10.1002/anie.201501742, 1038/5733, 10.1002/anie.201409679,)
REPLY: Thank you for the references, especially the one on mAChRs, which has implications for our ORs studies. Regarding the other references, although not directly related to the present paper, we do recognize the importance of water and ions in the behavior of GPCRs, and we are conducting separate studies on such effects. The message in this paper is that the “deep binding site” contains structural elements which, if adequately perturbed, can strongly modulate the properties already imparted by the body of the ligand. We had already hinted at this in [8] but have now elaborated.
ii. The authors do not compare the binding modes of the described compounds to the binding poses present in available opioid receptor crystal structures.
REPLY: Each ligand is different, so only general features can be discussed (cf. response to referee #2). Basic commonalities with cocrystallized ligands have been used to define the initial binding modes (see Methods), and these features are not lost during the simulations. These include the H-bond interactions of the charged amine and the overlay of the fused-ring scaffold (e.g., seen in the antagonist ß-FNA and naltrindole bound to the inactive MOR and DOR, respectively; cf. Methods). Other features depend on the specific chemistry of the compound, so a detailed discussion is outside the scope of this paper.
iii. N-phenethyl substituent is present in fentanyl structure. Could the authors compare positioning of N-phenethyl in their structures with that of this substituent in fentanyl? There are some recent models of fentanyl binding to mOR (1007/s00894-019-3999-2, 10.3390/molecules24040740) and some older ones (10.1021/jm9903702)
REPLY: The interactions are quite similar (for example, compare our Fig. S2 with Fig. 4 in 1007/s00894-019-3999-2). This should not be surprising given that the ring enters a deep, crowded pocket in both cases. The devil is in the details, though: the frequency and strength of contacts, and their correlations, are all critical factors that ultimately determine the activity and pharmacological outcome. The fentanyl body is also more flexible than our compounds, complicating the comparison.
iv. The authors claim that they equilibrated the simulations. Please provide the RMSD plots of the receptor helices (backbone of the helices) and of the ligand position (body and tail, in separate). Please add them to SI.
REPLY: By equilibration, we mean thermal equilibration (see Methods for details). Note that the RMSD of any part of a system provides no information on the equilibration of the whole. Regarding the structural relaxation of the system, all the calculations were performed during the late stages of dynamics (Dt), i.e., after the Ca-RMSDs were stabilized (typical behavior is shown below, Dt depends on the simulation; adding all the plots in the SI is unnecessary). A different, more meaningful criterion was used to assess the stability of the ligand, namely, the preservation of the -NH--D interactions (cf. Table S2).
v. The authors write that some contacts were frequent etc. Please provide the frequency plots, occupancy of the contacts.
REPLY: We added the frequency of some contacts in the SI and discussed the quantitative criteria used for the calculations (cf. Table S2 and text).
vi. The authors claim that “iii) Relative affinities correlate with the number and frequency of favorable contacts (polar/nonpolar, H-/halogen-bonding) between the substituent group and the OR; other contacts (body- and phenethyl ring-OR) are common to all the ligands/conformers and thus less critical in modulating the OR behavior.” Is this a quantitative correlation or merely a qualitative “opinion”? Please provide a correlational plot and the statistics of correlation.
REPLY: As stated, all the discussion in the SI is based on qualitative observations of the data, which we hope may help chemists design related compounds with new properties. The correlation mentioned is not a mathematical one (although maybe one can be derived).
vii. Please analyse the dihedral angles (means +/- S.D. taken from a few repetitions) of binding site residues’s side chains contacting the ligands. Are there any (quantitative) correlations with the experimental affinity/activity? With the substituent volumes? You could look at these examples: e.g. 3390/molecules24040740, here the authors found correlations between W6.48 dihedral angles and the volume of N-substituent in fentanyls. A correlation between W6.48 dihedrals and the functional activity was suggested in 10.1016/j.jmb.2017.05.009. Do the authors have similar relationships in their data? Finding such and other correlations (or lack thereof) would enable the authors to include results for mor than just 2e and 2i.
REPLY: Indeed, once the dynamic trajectories are available, many kinds of analyses can be done, some more relevant, some more sophisticated than others. Unfortunately, all require time and effort, which must be weighted vis-à-vis the scope and interest of the present paper. Here, we performed the calculations that we deem sufficient to understand the experimental data and help synthesize N-phenethyl-based compounds with better properties. That said, we do appreciate the suggestions, which gave us ideas to explore in a separate theoretical study.
viii. The AA numbering in Figure 1 and Figure 2 and S3 is at least partially incorrect. think that in mOR the aspartic acid anchoring the amine is D147 (in murine/rat sequence), W of the TM6 might be W293 etc. etc. Please check this carefully! You can consider using the Ballestros-Weinstein numbering, this would facilitate comparisons between mOR and dOR.
REPLY: We have renumbered the sequence for easier comparison with other studies and the crystal structures. In our experience, the relative numbering scheme of BW (proposed in the early ‘90s) is becoming less practical after the OR structures became available (from 2011 onward), so we prefer to avoid it.
ix. Figure S1A should have some captions allowing the reader to have some orientation ‘at which depth’ these blue, red, yellow green etc. residues are present. Please mark e.g. D3.32, W6.48 or alike to provide orientation for the reader. Which receptor is shown here? Which structure.
REPLY: It is now clarified.
7. Minor (‘editorial’) issues.
a. The manuscript is divided in a strange manner. There is a section ‘2. Results and Discussion’ and a section “3. Biological Results and Discussion”. This is unusual and as per my reading of MDPI author guidelines it is not allowed. Further, modelling is given as “subsection” of “in vitro studies” which is strange.
REPLY: Changed section 3 to section 2.3. In Vivo data
b. Some parts of the text seem to be placed in the wrong place. For example:
i. In the ‘Chemistry’ subsection, the authors describe the preparation of S10, then they talk about what novel compounds they report in this contribution. Then all of a sudden they talk about binding data, or functional assays. That these assays were done and their results are in Tables this and that, should start the subsection that follows.
REPLY: This has been done, as suggested.
ii. I cannot understand why 2a-2i and 2j/2k are separated in “chemistry”, there seem not anything ‘chemically’ special about them. That they were tested for cAMP accumulation and not for binding/GTPgS, should be explained in the sections that follow.
REPLY: This has been done, as suggested.
iii. By the way – why was it so?
Reply: Initial in vitro data were obtained before the decision was made to examine what the bias factors of the compounds might be. At that time the GTP assay lab was not set up for bias factor assays. We then moved to the cAMP assay lab for the bias factor assays, and decided it would be of interest to compare the two functional assays. We then re-ran most of the assays using the forskolin-induced cAMP accumulation assay and the PathHunter arrestin assay, adding a few compounds that were missing from the GTP assays.
iv. The last paragraph of 2.2.2. seems (perhaps) better placed at the beginning 2.2.3.
REPLY: This has been done, as suggested
v. Why are the results of the antinociceptive test given under 3.2. respiratory depression studies? Ought there not to be a separate subsection for them?
REPLY: The antinociceptive material was placed under the respiratory depression studies in squirrel monkeys because those assays were run prior to the respiratory depression assays, and the doses for that assay were dependent on the antinociceptive assays. Further, Table 3 contains data from the antinociceptive assays and the respiratory depression assays. If we put Table 3 in a new respiratory depression subsection, we would have antinociception there as well. We did consider having an antinociceptive subsection initially, but then decided it was simpler, and clearer, to leave it as written. We do not believe that it will cause great difficulty for the reader to understand these data if we combine the subsections in this way. To help in that regard, we have renamed section 2.3.2 Antinoiceptive Studies and Respiratory Depression Studies in Squirrel Monkeys.
c. P2. (the 2nd paragraph). “Tolerance[1-4]” -> please add space.
REPLY: This has been done, as suggested
d. P3. (last paragraph of introduction) – I am not English native speaker (and many of the authors seem to be), but it seems to me that ‘the question of what… has not hitherto been ascertained’ is not correct. I do not think one can ‘ascertain a question’, but I may be wrong, please check this.
REPLY: We thank the reviewer for the correction. We have modified the sentence to read: “The question of what that “correct” delta/mu ratio might be has not as yet been answered.”
e. P3. (bottom of the page) I think “secondary amine 10” should be “secondary amine S10”.
REPLY: This has been done, as suggested
page 4, Scheme 2.
i. The compounds should be labelled "1a-d" and "2a-k" so that it is clear that there are more than one as a result of a given reaction series.
REPLY: This has been done, as suggested
ii. As to the bottom part: it seems that the usage of "R2" is inconsistent. With the reaction arrow, it denotes the alkyl chain to which Br atom is attached. With the structure 2 (bottom, right) and the box with red borders, it denotes a substitution pattern at the N-phenethyl ring. To me, it seems inconsistent. Please correct.
REPLY: This has been done. R2Br was changed to R2PhCH2CH2Br
iii. There are two dots after the Scheme caption.
REPLY: dot removed
f. P5. The table caption contains some strange characters instead of mu, delta, kappa and gamma. This is perhaps only the issue of the pdf generated by the submission system, but please check this to be sure.
REPLY: The Table caption appeared to be ok when we opened the zip file. Hopefully, it will remain that way.
g. P6 (last paragraph): Ki = < 1 nM should rather read Ki < 1 nM
REPLY: changed all to Ki <
h. page 17, synthesis
i. Why 2i is described before 1a, 1b, 2a… etc? Please consider putting this after 2h and before
REPLY Reply: 2i is the example used for the general procedure - the procedure used for the other compounds. We thought it would be good to use 2i for the complete procedural description because, for us, it was the most important compound in the manuscript. However, to make that clearer, we have added the IUPAC name for 2i and added: “See the “General Procedure in 3.2.” after 2h and before 2j, so anyone looking for it in that area would now see that it is the General Procedure.
2i. HCl is described in a separate paragraph while obtaining of other HCl salts is given together with their respective bases. Please make this consistent.
REPLY: This has been done.
ii. Please make 2i, 1a, 1b, 1c, 1d, 2a, 2b, 2c, 2d, 2e, 2f, 2g in paragraph headings in bold.
REPLY: This has been done
i. EC50 is written without subscript. On the other hand ED50 is written with subscript. This should be consistent. I think the subscript is these days more common, so my advice would be to use it here. At least, please, make the usage consistent (there are many instances of EC50 in the text, I shall not enumerate them here). Similarly, I would personally prefer Ki with ‘i' in subscript.
REPLY: Since there were far fewer ED50s, we changed them all to unsubscripted. Ki was modified with ‘i' in subscript
j. Page 22. 4.3. “contained100 uL” -> please add space.
REPLY: done.
k. Page 22. Why is 4.3 “Pharmacology. Opioid …” not under ‘In vitro pharmacology” and other assays are?
REPLY: This has been done. Numbering modified.
l. SI, NMRs. The NMRs are not numbered. It is a little inconvenient for the reader that they have no textual ‘description’ like ‘2a’, ‘2b’ etc. Please add at least such a label.
REPLY: This has been done. Now labeled ( e.g., "1H and 13C NMR of 2a").
m. SI, NMRs. The structures on p. 19 and 20 (S11 derivative) have different stereochemistries than S11 in Table S1, Table 2 and all other derivatives in body and SI. Please check this and correct.
REPLY: They have been corrected.
n. SI, Scheme S1. Please note in the figure caption that the percent values are yields, are they? In the lower part, what was the yield for S10 to S11 conversion? I think “10 ~ 70 %” applies to the diverse derivatives 2a-j discussed in the body, but not to S11. Please correct.
REPLY:: Corrected – removed 10-70%. Scheme’s title now reads: “Scheme S1. Synthesis of Hydromorphone (S5), N-Norhydromorphone (S10) and N-Phenethylnorhydromorphone (S11). Reaction Conditions and Yield (%)”
o. SI, Figure S2. The Figure caption says that the maps were constructed with ‘B3LYP/DGDZVP’. Were the structures optimized? Please note this.
REPLY: The electrostatic potential maps were constructed based on the optimized structures. The figure caption has been corrected.
Round 2
Reviewer 3 Report
The authors did substantial effort to improve the manuscript following my remarks. Although they did not apply to all of them (regarding the modelling, in particular), their explanations are convincing, and while I would do some things here the other way, their way can be acceptable.
Therefore, I recommend the manuscript be accepted in the present form.